# MISSINGNESS BIAS IN MODEL DEBUGGING

**Saachi Jain**[1*], **Hadi Salman**[1*], **Eric Wong**[1], **Pengchuan Zhang**[2],
**Vibhav Vineet**[2], **Sai Vemprala**[2], **Aleksander Mądry**[1]

[1]Massachusetts Institute of Technology
[2]Microsoft Research
[1]{saachij, hady, wongeric, madry}@mit.edu
[2]{penzhan, vivineet, sai.vemprala}@microsoft.com

## ABSTRACT

Missingness, or the absence of features from an input, is a concept fundamental to many model debugging tools. However, in computer vision, pixels cannot simply be removed from an image. One thus tends to resort to heuristics such as blacking out pixels, which may in turn introduce bias into the debugging process. We study such biases and, in particular, show how transformer-based architectures can enable a more natural implementation of missingness, which side-steps these issues and improves the reliability of model debugging in practice.[1]

## 1 INTRODUCTION

Model debugging aims to diagnose a model's failures. For example, researchers can identify global biases of models via the extraction of human-aligned concepts (Bau et al., 2017; Wong et al., 2021), or understand the texture bias by analyzing the models performance on synthetic datasets (Geirhos et al., 2019; Leclerc et al., 2021). Other approaches aim to highlight local features to debug individual model predictions (Simonyan et al., 2013; Dhurandhar et al., 2018; Ribeiro et al., 2016a; Goyal et al., 2019).

A common theme in these methods is to compare the behavior of the model *with and without* certain individual features (Ribeiro et al., 2016a; Goyal et al., 2019; Fong & Vedaldi, 2017; Dabkowski & Gal, 2017; Zintgraf et al., 2017; Dhurandhar et al., 2018; Chang et al., 2019). For example, interpretability methods such as LIME (Ribeiro et al., 2016b) and integrated gradients (Sundararajan et al., 2017) use the predictions when certain features are removed from the input to attribute different regions of the input to the decision of the model. Dhurandhar et al. (2018) find minimal regions in radiology images that are necessary for classifying a person as having autism. Fong & Vedaldi (2017) propose learning image masks that minimize a class score to achieve interpretable explanations. Similarly, in natural language processing, model designers often remove individual words to understand their importance to the output (Mardaoui & Garreau, 2021; Li et al., 2016). The absence of features from an input, a concept sometimes referred to as *missingness* (Sturmfels et al., 2020), is thus fundamental to many debugging tools.

However, there is a problem: while we can easily remove words from sentences, removing objects from images is not as straightforward. Indeed, removing a feature from an image usually requires approximating missingness by replacing those pixel values with something else, e.g., black color. However, these approximations tend not to be perfect (Sturmfels et al., 2020). Our goal is thus to give a holistic understanding of missingness and, specifically, to answer the question:

*How do missingness approximations affect our ability to debug ML models?*

---

[*]Equal contribution.
[1]Our code is available at https://github.com/madrylab/missingness.

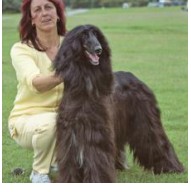
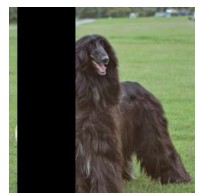
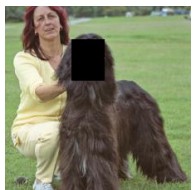

(a) Original image        (b) Masking the human        (c) Masking the dog's snout

Figure 1: Consider an image of a dog being held by its owner. By removing the owner from the image, we can study how much our model's prediction depends on the presence of a human. In a similar vein, we can identify which aspects of the dog (head, body, paws) are most critical for classifying the image by ablating these parts.

OUR CONTRIBUTIONS

In this paper, we investigate how current missingness approximations, such as blacking out pixels, can result in what we call *missingness bias*. This bias turns out to hinder our ability to debug models. We then show how transformer-based architectures can enable a more natural implementation of missingness, allowing us to side-step this bias. More specifically, our contributions include:

**Pinpointing the missingness bias.** We demonstrate at multiple granularities how simple approximations, such as blacking out pixels, can lead to missingness bias. This bias skews the overall output distribution toward unrelated classes, disrupts individual predictions, and hinders the model's use of the remaining (unmasked) parts of the image.

**Studying the impact of missingness bias on model debugging.** We show that missingness bias negatively impacts the performance of debugging tools. Using LIME—a common feature attribution method that relies on missingness—as a case study, we find that this bias causes the corresponding explanations to be inconsistent and indistinguishable from random explanations.

**Using vision transformers to implement a more natural form of missingness.** The token-centric nature of vision transformers (ViT) (Dosovitskiy et al., 2021) facilitates a more natural implementation of missingness: simply drop the corresponding tokens of the image subregion we want to remove. We show that this simple property substantially mitigates missingness bias and thus enables better model debugging.

## 2 MISSINGNESS

Removing features from the input is an intuitive way to understand how a system behaves (Sturmfels et al., 2020). Indeed, by comparing the system's output with and without specific features, we can infer what parts of the input led to a specific outcome (Sundararajan et al., 2017)—see Figure 1. The absence of features from an input is sometimes referred to as *missingness* (Sturmfels et al., 2020).

The concept of missingness is commonly leveraged in machine learning, especially for tasks such as model debugging. For example, several methods for feature attribution quantify feature importance by studying how the model behaves when those features are removed (Sturmfels et al., 2020; Sundararajan et al., 2017; Ancona et al., 2017). One commonly used method, LIME (Ribeiro et al., 2016a), iteratively turns image subregions on and off in order to highlight its important parts. Similarly, integrated gradients (Sundararajan et al., 2017), a typical method for generating saliency maps, leverages a "baseline image" to represent the "absence" of features in the input. Missingness-based tools are also often used in domains such as natural language processing (Mardaoui & Garreau, 2021; Li et al., 2016) and radiology (Dhurandhar et al., 2018).

**Challenges of approximating missingness in computer vision.** While ignoring parts of an image is simple for humans, removing image features is far more challenging for computer vision models (Sturmfels et al., 2020). After all, convolutional networks require a structurally contiguous image as an input. We thus cannot leave a "hole" in the image where the model should ignore the input.

| Original | Random | Least Salient | Most Salient |
|---|---|---|---|
| 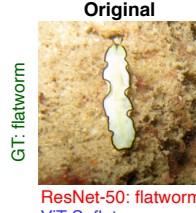 | 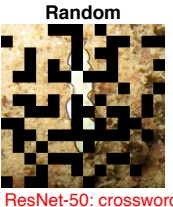 | 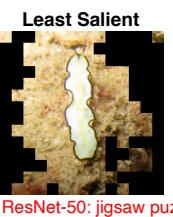 | 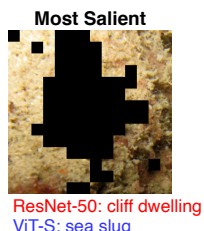 |

GT: flatworm

ResNet-50: flatworm / ViT-S: flatworm  ResNet-50: crossword / ViT-S: flatworm  ResNet-50: jigsaw puzzle / ViT-S: flatworm  ResNet-50: cliff dwelling / ViT-S: sea slug

Figure 2: Given an image of a `flatworm`, we remove various regions of the original image; masking for ResNet, and dropping tokens for ViT. **(Section 2.1):** Irrespective of what subregions of the image are removed (least salient, most salient, or random), a ResNet-50 outputs the wrong class (`crossword`, `jigsaw puzzle`, `cliff dwelling`). Taking a closer look at the randomly masked image of Figure 2, we notice that the predicted class (`crossword puzzle`) is not totally unreasonable given the masking pattern. The model seems to be relying on the masking pattern to make the prediction, rather than the remaining (unmasked) portions of the image. **(Section 2.2):** The ViT-S on the other hand either maintains its original prediction or predicts a reasonable label given remaining image subregions.

Consequently, practitioners typically resort to approximating missingness by replacing these pixels with other, intended to be "meaningless", pixels.

Common *missingness approximations* include replacing the region of the image with black color, a random color, random noise, a blurred version of the region, and so forth (Sturmfels et al., 2020; Ancona et al., 2017; Smilkov et al., 2017; Fong & Vedaldi, 2017; Zeiler & Fergus, 2014; Sundararajan et al., 2017). However, there is no clear justification for why any of these choices is a good approximation of missingness. For example, blacked out pixels are an especially popular baseline, motivated by the implicit heuristic that near zero inputs are somehow neutral for a simple model (Ancona et al., 2017). However, if only part of the input is masked or the model includes additive bias terms, the choice of black is still quite arbitrary. In (Sturmfels et al., 2020), the authors found that saliency maps generated with integrated gradients are quite sensitive to the chosen baseline color, and thus can change significantly based on the (arbitrary) choice of missingness approximation.

## 2.1 MISSINGNESS BIAS

What impact do these various missingness approximations have on our models? We find that current approximations can cause significant bias in the model's predictions. This causes the model to make errors based on the "missing" regions rather than the remaining image features, rendering the masked image out-of-distribution.

Figure 2 depicts an example of these problems. If we mask a small portion of the image, irrespective of which part of the image that is, convolutional networks (CNNs) output the wrong class. In fact, CNNs seem to be relying on the masking pattern to make the prediction, rather than the remaining (unmasked) portions of the image. This type of behavior can be especially problematic for model debugging techniques, such as LIME, that rely on removing image subregions to assign importance to input features. Further examples can be found in Appendix C.1.

There seems to be an inherent bias accompanying missingness approximations, which we refer to as the *missingness bias*. In Section 3, we systematically study how missingness bias can affect model predictions at multiple granularities. Then in Section 4, we find that missingness bias can cause undesirable effects when using LIME by causing its explanations to be inconsistent and indistinguishable from random explanations.

## 2.2 A MORE NATURAL FORM OF MISSINGNESS VIA VISION TRANSFORMERS

The challenges of missingness bias raises an important question: what constitutes a correct notion of missingness? Since masking pixels creates biases in our predictions, we would ideally like to remove those regions from consideration entirely. Because convolutional networks slide filters across the

image, they require spatially contiguous input images. We are thus limited to replacing pixels with some baseline value (such as blacking out the pixels), which leads to missingness bias.

Vision transformers (ViTs) (Dosovitskiy et al., 2021) use layers of self-attention instead of convolutions to process the image. Attention allows the network to focus on specific sub-regions while ignoring other parts of the input (Vaswani et al., 2017; Xu et al., 2015); this allows ViTs to be more robust to occlusions and perturbations (Naseer et al., 2021). These aspects make ViTs especially appealing for countering missingness bias in model debugging.

In particular, we can leverage the unique properties of ViTs to enable a far more natural implementation of missingness. Unlike CNNs, ViTs operate on sets of *image tokens*, each of which correspond to a positionally encoded region of the image. Thus, in order to remove a portion of the image, *we can simply drop the tokens that correspond to the regions of the image we want to "delete."* Instead of replacing the masked region with other pixel values, we can modify the forward pass of the ViT to directly remove the region entirely.

We will refer to this implementation of missingness as *dropping tokens* throughout the paper (see Appendix B for further details). As we will see, using ViTs to drop image subregions will allow us to side-step missingness bias (see Figure 2), and thus enable better model debugging[2].

## 3 THE IMPACTS OF MISSINGNESS BIAS

Section 2.1 featured several qualitative examples where missingness approximations affect the model's predictions. Can we get a precise grasp on the impact of such missingness bias? In this section, we pinpoint how missingness bias can manifest at several levels of granularity. We further demonstrate how, by enabling a more natural implementation of missingness through dropping tokens, ViTs can avoid this bias.

**Setup.** To systematically measure the impacts of missingness bias, we iteratively remove subregions from the input and analyze the types of mistakes that our models make. See Appendix A for experimental details. We perform an extensive study across various: architectures (Appendix C.3), missingness approximations (Appendix C.4), subregion sizes (Appendix C.5), subregion shapes: patches vs superpixels (Appendix C.6), and datasets (Appendix E).

Here we present our findings on a single representative setting: removing $16 \times 16$ patches from ImageNet images through blacking out (ResNet-50) and dropping tokens (ViT-S). The other settings lead to similar conclusions as shown in Appendix C. Our assessment of missingness bias, from the overall class distribution to individual examples, is guided by the following questions:

**To what extent do missingness approximations skew the model's overall class distribution?** We find that missingness bias affects the model's overall class distribution (i.e the probability of predicting any one class). In Figure 3, we measure the shift in the model's output class distribution before and after image subregions are randomly removed. The overall entropy of output class distribution degrades severely. In contrast, this bias is eliminated when dropping tokens with the ViT. The ViT thus maintains a high class entropy corresponding to a roughly uniform class distribution. These findings hold regardless of what order we remove the image patches (see Appendix C.2).

**Does removing random or unimportant regions flip the model's predictions?** We now take closer look at how missingness approximations can affect individual predictions. In Figure 4, we plot the fraction of examples where removing a portion of the image flips the model's prediction. We find that the ResNet rapidly flips its predictions even when the less relevant regions are removed first. This degradation is thus more likely due to missingness bias rather than the removal of individual regions. In contrast, the ViT maintains its original predictions even when large parts of the image are removed.

**Do remaining unmasked regions produce reasonable predictions?** When removing regions of the image with missingness, we would hope that the model makes a "best-effort" prediction

---

[2]Unless otherwise specified, we drop tokens for the vision transformers when analyzing missingness bias on ViTs. An analysis of the missingness bias for ViTs when blacking out pixels can be found in Appendix C.7.

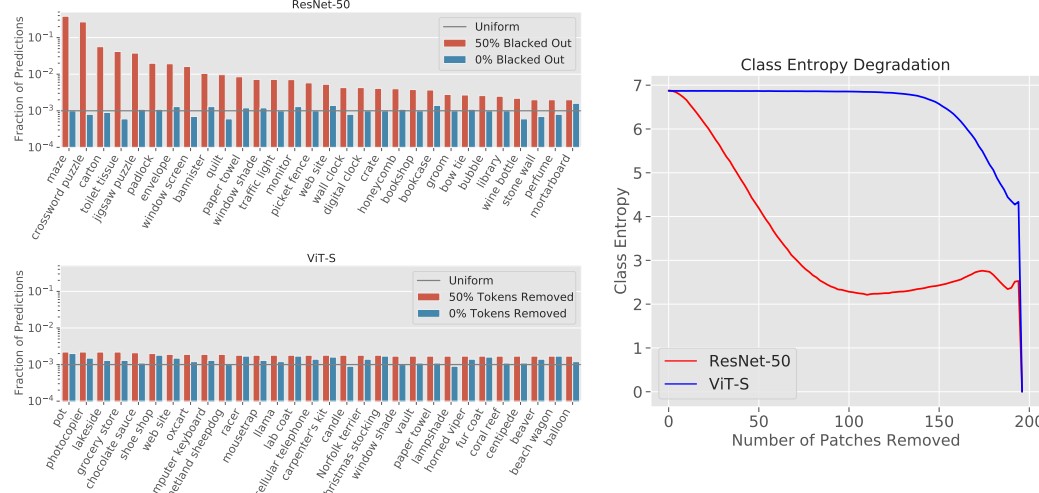

Figure 3: We measure the shift in output class distribution after applying missingness approximations. **Left**: Fraction of images predicted as each class (*on a log scale*) before and after randomly removing 50% of the image. We display the most frequently predicted 30 classes after applying the missingness approximations. **Right**: Degradation in overall class entropy as subregions are removed. As patches are blacked out, the ResNet's predictions skew from a uniform distribution toward a few unrelated classes such as maze, crossword puzzle, and carton. On the other hand, the ViT maintains a uniform class distribution with high class entropy.

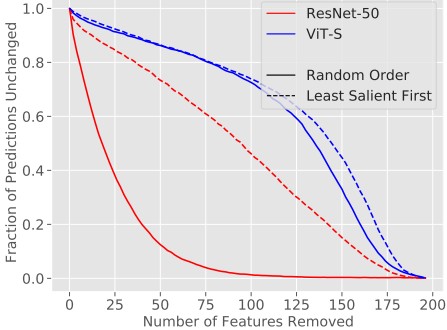
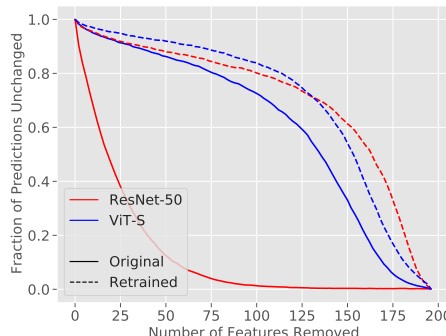

Figure 4: We plot the fraction of images where the prediction does not change as image regions are removed. The ResNet flips its predictions even when unrelated patches are removed, while the ViT maintains its original prediction.

Figure 5: We repeat the experiment in Figure 4 with models retrained with missingness augmentations. Applying missingness approximations during training mitigates missingness bias for ResNets.

given the remaining image features. This assumption is critical for interpretability methods such as LIME (Ribeiro et al., 2016a), where crucial features are identified by iteratively masking out image subregions and tracking the model's predictions.

Are our models actually using the remaining uncovered features after missingness approximations are applied though? To answer this question, we measure how semantically related the model's predictions are after masking compared to its original prediction using a similarity metric on the WordNet Hierarchy (Miller, 1995) as shown in Figure 6. By the time we mask out 25% of the image, the predictions of the ResNet largely become irrelevant to the input. ViTs on the other hand continue to predict classes that are related to the original prediction. This indicates that ViTs successfully leverage the remaining features in the image to provide a reasonable prediction.

**Can we remove missingness bias by augmenting with missingness approximations?** One way to remove missingness bias could be to apply missingness approximations during training. For

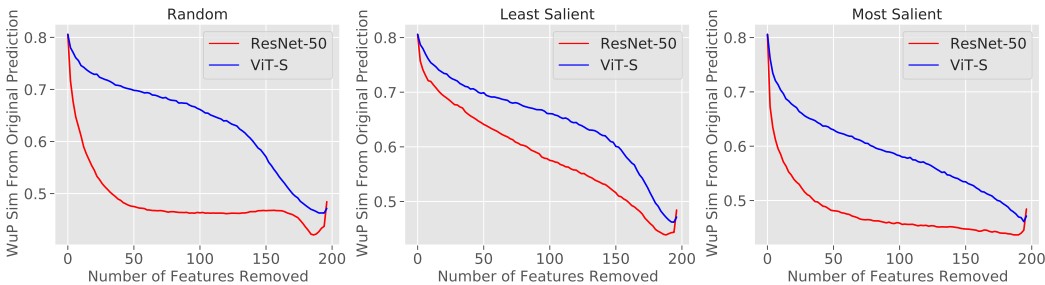

Figure 6: We iteratively remove image regions in the order of random, most salient, and least salient. We then plot the average WordNet similarity between the original prediction and the new prediction if the predictions differ. We find that ViT-S, even when the prediction changes, continues to predict something relevant to the original image.

example, in RemOve and Retrain (ROAR), Hooker et al. (2018) suggest retraining multiple copies of the model by blacking out pixels during training (see Appendix F for an overview on ROAR).

To check if this indeed helps side-step the missingness bias, we retrain our models by randomly removing 50% of the patches during training, and again measure the fraction of examples where removing image patches flips the model's prediction (see Figure 5). While there is a significant gap in behavior between the standard and retrained CNNs, the ViT behaves largely the same. This result indicates that, while retraining is important when analyzing CNNs, it is unnecessary for ViTs when dropping the removed tokens: we can instead perform missingness approximations directly on the original model while avoiding missingness bias for free. See Appendix F for more details.

## 4 MISSINGNESS BIAS IN PRACTICE: A CASE STUDY ON LIME

Missingness approximations play a key role in several feature attribution methods. One attribution method that fundamentally relies on missingness is the local interpretable model-agnostic explanations (LIME) method (Ribeiro et al., 2016a). LIME assigns a score to each image subregion based on its relevance to the model's prediction. Subregions of the image with the top scores are referred to as *LIME explanations*. A crucial step of LIME is "turning off" image subregions, usually by replacing them with some baseline pixel color. However, as we found in Section 2, missingness approximations can cause missingness bias, which can impact the generated LIME explanations.

We thus study how this bias impacts model debugging with LIME. To this end, we first show that missingness bias can create inconsistencies in LIME explanations, and further cause them to be indistinguishable from random explanations. In contrast, by dropping tokens with ViTs, we can side-step missingness bias in order to avoid these issues, enabling better model debugging.

Figure 7 depicts an example of LIME explanations. Qualitatively, we note that explanations generated for standard ResNets seem to be less aligned with human intuition than ViTs or ResNets retrained with missingness augmentations[3].

**Missingness bias creates inconsistent explanations.** Since LIME uses missingness approximations while scoring each image subregion, the generated explanations can change depending on which approximation is used. How consistent are the resulting explanations? We generate such explanations for a ViT and a CNN using 8 different baseline colors. Then, for each pair of colors, we measure how much their top-k features agree (see Figure 8). We find that the ResNet produces explanations that are almost as inconsistent as randomly generated explanations. The explanations of the ViT, however, are always consistent by construction since the ViT drops tokens entirely. For further comparison, we also plot the consistency of the LIME explanations of a ViT-S where we mask out pixels instead of drop the tokens.

**Missingness bias renders different LIME explanations indistinguishable.** Do LIME explanations actually reflect the model's predictions? A common approach to answer this is to remove the

---

[3]See Appendix D.1 for more details on this. We also include an overview of LIME and detailed experimental setup for this section in Appendix A, and further experiments using superpixels in Appendix D.2.

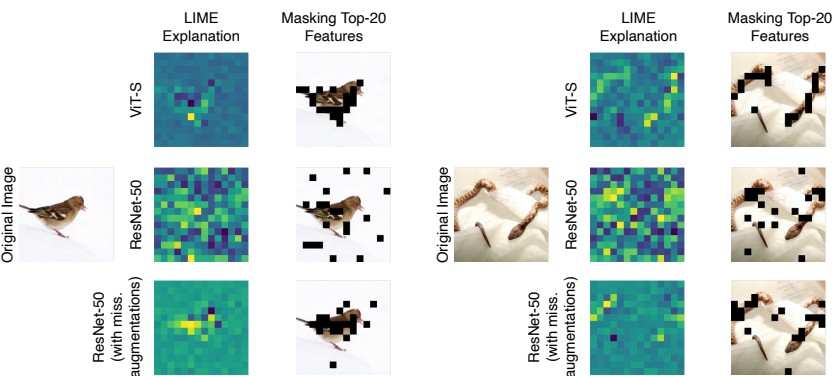

Figure 7: Examples of generated LIME explanations and masking the top 20 features. Since LIME requires removing image features, it can be subject to missingness bias. We note that LIME explanations generated for standard ResNets seem to be less aligned with human intuition than ViTs or ResNets retrained with missingness augmentations (See Appendix D.1 for more examples).

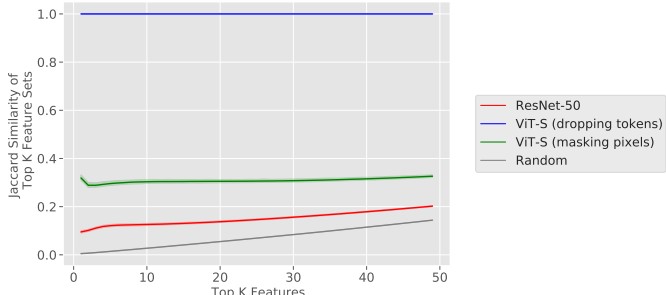

Figure 8: We plot the agreement (using Jaccard similarity) of top-k features across LIME explanations of 28 pairs of baseline colors. The result is averaged over the 28 pairs, and we display the 95% confidence interval over the pairs of colors. ResNet-50's explanations are almost as consistent as random explanations. For ViT with dropping tokens, explanations are naturally always consistent.

top-k subregions (by masking using a missingness approximation), and then check if the model's prediction changes (Samek et al., 2016). This is sometimes referred to as the *top-K ablation test* (Sturmfels et al., 2020). Intuitively, an explanation is better if it causes the predictions to flip more rapidly. We apply the top-k ablation test of four different LIME explanations on a ResNet-50 and a ViT-S as shown in Figure 9. Specifically, for each model we evaluate: 1) its own generated explanations, 2) the explanations of an identical architecture trained with a different seed 3) the explanations of the other architecture and 4) randomly generated explanations.

For CNNs (Figure 9-left), all four explanations (even the random one) flip the predictions at roughly an equal rate in the top-K ablation test. In these cases, the bias incurred during evaluation plays a larger role in changing the predictions than the importance of the region being removed, rendering the four explanations indistinguishable. On the ViT however (Figure 9-right), the LIME explanation generated from the original model outperforms all other explanations (followed by an identical model trained with a different seed). As we would expect, the ResNet and the random explanations cause minimal prediction flipping, which indicates that these explanations do not accurately capture the feature importance for the ViT. Thus, unlike for the CNNs, the different LIME explanations for the ViT are distinguishable from random (and quantitatively better) via the top-k ablation test.

**What happens if we retrain our models with missingness augmentations?** As in Section 3, we repeat the above experiment on models where 50% of the patches are removed during training. The results are reported in Figure 10. We find that the LIME explanations evaluated with the retrained CNN are now distinguishable, and the explanation generated by the same CNN outperforms the other explanations. Thus, retraining with missingness augmentation "fixes" the CNN and makes the

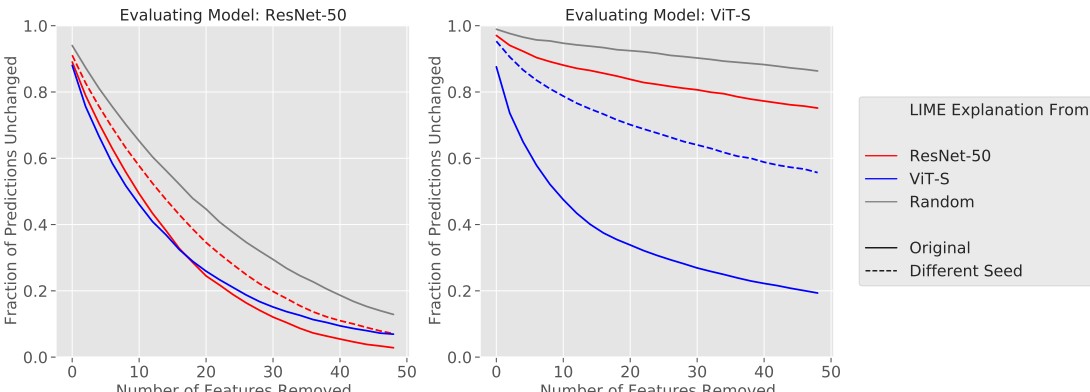

Figure 9: We evaluate LIME explanations using the top-K ablation test on a ResNet and ViT by measuring the fraction of examples who keep their original prediction after removing the Top-K features. A sharper degradation indicates a more appropriate explanation for that model. While the LIME scores on the ResNet are largely indistinguishable, the ViT shows clear differentiation between the different explanations.

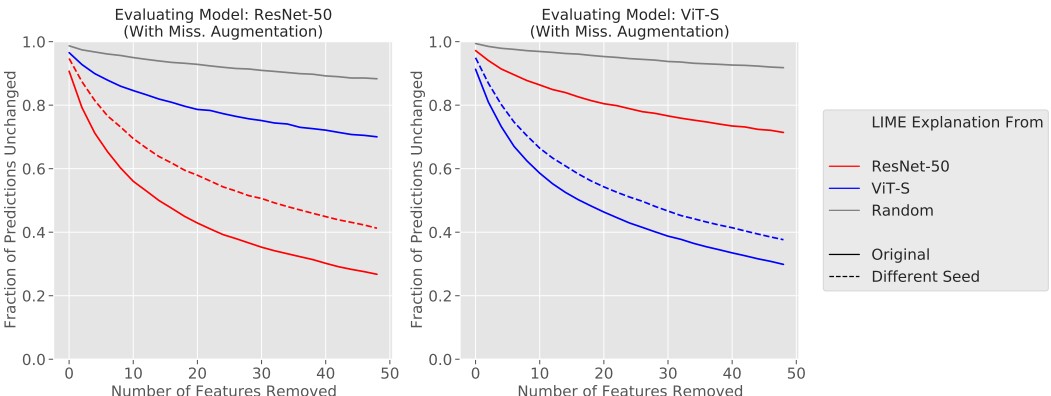

Figure 10: We replicate the experiment in Figure 9, but instead use models where missingness approximations were introduced during training. This procedure fixes evaluation issues for ResNets, but does not substantially change the evaluation picture of ViTs.

top-k ablation test more effective by mitigating missingness bias. On the other hand, since the ViT already side-steps missingness bias by dropping tokens, the top-k ablation test does not substantially change when using the retrained model. We can thus evaluate LIME explanations directly on the original ViT without resorting to surrogate models.

## 5 RELATED WORK

**Interpretability and model debugging.** Model debugging is a common goal in interpretability research where researchers try to justify model decisions based on either local features (i.e., specific to a given input) or global ones (i.e., general biases of the model). Global interpretability methods include concept-based explanations (Bau et al., 2017; Kim et al., 2018; Yeh et al., 2020; Wong et al., 2021). They also encompass many of the robustness studies including adversarial (Szegedy et al., 2014; Goodfellow et al., 2015; Madry et al., 2018) or synthetic perturbations (Geirhos et al., 2019; Engstrom et al., 2019; Kang et al., 2019; Hendrycks & Dietterich, 2019; Xiao et al., 2020; Zhu et al., 2017) which can be cast as uncovering global biases of vision models (e.g. the model is biased to textures (Geirhos et al., 2019)). Local explanation methods, also known as *feature attribution methods*, aim to highlight important regions in the image causing the model prediction. Many of these methods use gradients to generate visual explanations, such as saliency maps (Simonyan et al., 2013; Dabkowski & Gal, 2017; Sundararajan et al., 2017). Others explain the model predictions by studying the behavior of models *with and without* certain individual features (Ribeiro et al., 2016a;

Goyal et al., 2019; Fong & Vedaldi, 2017; Dabkowski & Gal, 2017; Zintgraf et al., 2017; Dhurandhar et al., 2018; Chang et al., 2019; Hendricks et al., 2018; Singla et al., 2021). They focus on the change in classifier outputs with respect to images where some parts are masked and replaced with various references such as random noise, mean pixel values, blur, outputs of generative models, etc.

**Evaluating feature attribution methods.** It is important for feature attribution methods to truly reflect why the model made a decision. Unfortunately, evaluating this is hard since we lack the ground truth of what parts of the input are important. Several works showed that visual assessment fails to evaluate attribution methods (Adebayo et al., 2018; Hooker et al., 2018; Kindermans et al., 2017; Lin et al., 2019; Yeh et al., 2019; Yang & Kim, 2019; Narayanan et al., 2018), and instead proposed several qualitative tests as replacements. Samek et al. (2016) proposed the region perturbation method which removes pixels according to the ranking provided by the attribution maps, and measures how the prediction changes i.e., how the class encoded in the image disappears when we progressively remove information from the image. Later on, Hooker et al. (2018) proposed remove and retrain (ROAR) which showed that in order for region perturbation method to be more informative, the model has to be trained with those perturbations. Kindermans et al. (2017) posit that feature attribution methods should fulfill invariance with respect to some set of transformations, e.g. adding a constant shift to the input data. Adebayo et al. (2018) proposed several sanity checks that feature attribution methods should pass. For example, a feature attribution method should produce different attributions when evaluated on a trained model and a randomly initialized model. Zhou et al. (2021) proposed a modifying datasets such that a model must rely on a set of known and well-defined features to achieve high performance, thus offering a ground truth for feature attribution methods.

**The notion of missingness.** The notion of missingness is commonly used in machine learning, especially for tasks such as feature attribution (Sturmfels et al., 2020; Sundararajan et al., 2017; Hooker et al., 2018; Ancona et al., 2017). Practitioners leverage missingness in order to quantify the importance of specific features, by evaluating the model when removing the feature in the input (Ribeiro et al., 2016a; Goyal et al., 2019; Fong & Vedaldi, 2017; Dabkowski & Gal, 2017; Zintgraf et al., 2017; Dhurandhar et al., 2018; Chang et al., 2019; Sundararajan et al., 2017; Carter et al., 2021; Covert et al., 2021). For example, in natural language processing, model designers often remove tokens to assign importance to individual words (Mardaoui & Garreau, 2021; Li et al., 2016). In computer vision, missingness is foundational to several interpretability methods. For example, LIME (Ribeiro et al., 2016a) iteratively turns image features on and off in order to learn the importance of each image subregion. Similarly, integrated gradients (Sundararajan et al., 2017) requires a "baseline image" that is used to represent "absence" of feature in the input. It is thus important to study missingness and how to properly approximate it for computer vision applications.

**Vision transformers.** Our work leverages the vision transformer (ViT) architecture, which was first proposed by Dosovitskiy et al. (2021) as a direct adaptation of the popular transformer architecture used in NLP applications (Vaswani et al., 2017) for computer vision. In particular, ViTs do not include any convolutions, instead they tokenize the image into patches which are then passed through several full layers of multi-headed self-attention. These help the transformer *globally share* information between all image regions at every layer. While convolutions must iteratively expand their receptive fields, vision transformers can immediately share information between patches throughout the entire network. ViTs recently got a lot of attention in the research community from works ranging from efficient methods for training ViTs (Touvron et al., 2020) to studying the properties of ViTs (Shao et al., 2021; Mahmood et al., 2021; Naseer et al., 2021; Salman et al., 2021). The work of Naseer et al. (2021) is especially related to our work as it studies the robustness of ViTs to a number of input modifications including occlusions. They find that the overall accuracy of CNNs degrades more quickly than ViTs when large regions of the input is masked. These findings complement our intuition that ResNets can experience bias when pixels are replaced with a variety of missingness approximations.

## 6 CONCLUSION

In this paper, we investigate how current missingness approximations result in missingness bias. We also study how this bias interferes with our ability to debug models. We demonstrate how transformer-based architectures are one possible solution, as they enable a more natural (and thus less biasing) implementation of missingness. Such architectures can indeed side-step missingness bias and are more reliable to debug in practice.

## 7 ACKNOWLEDGEMENTS

Work supported in part by the NSF grants CCF-1553428 and CNS-1815221. This material is based upon work supported by the Defense Advanced Research Projects Agency (DARPA) under Contract No. HR001120C0015.

Research was sponsored by the United States Air Force Research Laboratory and the United States Air Force Artificial Intelligence Accelerator and was accomplished under Cooperative Agreement Number FA8750-19-2-1000. The views and conclusions contained in this document are those of the authors and should not be interpreted as representing the official policies, either expressed or implied, of the United States Air Force or the U.S. Government. The U.S. Government is authorized to reproduce and distribute reprints for Government purposes notwithstanding any copyright notation herein.

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

## A  EXPERIMENTAL DETAILS.

### A.1  MODELS AND ARCHITECTURES

We use two sizes of vision transformers: ViT-Tiny (ViT-T) and ViT-Small (ViT-S) (Wightman, 2019; Dosovitskiy et al., 2021). We compare to residual networks of similar size: ResNet-18 and ResNet-50 (He et al., 2016), respectively. These architectures and their corresponding number of parameters are summarized in Table 1.

Table 1: A collection of neural network architectures we use in our paper.

| Architecture | ViT-T | ResNet-18 | ViT-S | ResNet-50 |
|---|---|---|---|---|
| Params | 5M | 12M | 22M | 26M |

### A.2  TRAINING DETAILS

We train our models on ImageNet (Russakovsky et al., 2015), with a custom (research, non-commercial) license, as found here `https://paperswithcode.com/dataset/imagenet`. For all experiments in this paper, we consider 10,000 image subset of the original ImageNet validation set (we take every 5th image).

1. For ResNets, we train using SGD with batch size of 512, momentum of 0.9, and weight decay of 1e-4. We train for 90 epochs with an initial learning rate of 0.1 that drops by a factor of 10 every 30 epochs.
2. For ViTs, we use the same training scheme as used in Wightman (2019).

Note that we use the same (basic) data-augmentation techniques for both ResNets and ViTs. Specifically, we only use random resized crop and random horizontal flip (no RandAug, CutMix, MixUp, etc.).

We attach all our model weights to the submission.

**Models trained with missingness augmentations.**  In Sections 3 and 4, we also consider models that were augmented with missingness approximations during training (inspired by ROAR (Hooker et al., 2018), see Appendix F for further discussion). We retrain our models by randomly removing 50% of the patches (by blacking out for ResNet and dropping the respective tokens for ViT). The other training hyperparameters are maintained the same as the standard models above.

**Infrastructure and computational time.**  For ImageNet, we train our models on 4 V100 GPUs each, and training took around 12 hours for ResNet-18 and ViT-T, and around 20 hours for ResNet-50 and ViT-S.

For CIFAR-10, we fine-tune pretrained ViTs and ResNets on a single V100 GPU. Fine-tuning ViT-T and ResNet-18 took around 1 hours, and fine-tuning ViT-S and ResNet-50 took around 1.5 hours.

All of our analysis can be run on a single 1080Ti GPU, where the time for one forward pass with batch size of 128 is reported in Table 2.

Table 2: A collection of neural network architectures we use in our paper.

| Architecture | ViT-T | ResNet-18 | ViT-S | ResNet-50 |
|---|---|---|---|---|
| Inference time (sec) | $0.031 \pm 0.018$ | $0.033 \pm 0.013$ | $0.041 \pm 0.016$ | $0.039 \pm 0.015$ |

### A.3  EXPERIMENTAL DETAILS FOR SECTION 3

For the experiments in Section 3, we iteratively remove subregions from the input. In the main paper, we consider removing $16 \times 16$ patches: we black out patches for the ResNet-50 and drop

the corresponding token for the ViT-S. We consider other patch sizes as well as superpixels in Appendix C.

We consider removing patches in three orders: random, most salient, and least salient. We use saliency as a rough heuristic for relevance to the image (typically, more salient regions tend to be in the foreground and less salient regions in the background). For all models, we determine the salience of an image subregion as the mean value of that subregion for a standard ResNet-50's saliency map (the order of patches removed is thus the same for both the ResNet and the ViT).

### A.4    EXPERIMENTAL DETAILS FOR SECTION 4

**Overview on LIME.**   Local interpretable model-agnostic explanations (LIME) Ribeiro et al. (2016b) is a common method for feature attribution. Specifically, LIME proceeds by generating perturbations of the image, where in each perturbation the subregions are randomly turned on or off. For ResNets, we turn off subregions by masking them with some baseline color, while for ViTs we drop the associated tokens. After evaluating these perturbations with the model, we fit classifier using Ridge Regression to predict the value of the logit of the original predicted class given the presence of each subregion. The LIME explanation is then the weight of each subregion in the ridge classifier (these are often referred to as LIME scores). We perform LIME with 1000 perturbations, and include an implementation of LIME in our attached code.

**Implementation details for LIME consistency plots.**   For the experiment in Figure 8, we evaluate LIME using 8 different baseline colors (the colors are generated by setting the R, G, and B values as either 0 or 1). Then, for each pair of colors, we measure the similarity of their top-k feature sets according to their LIME scores for varying k (using Jaccard similarity) averaged over 10,000 examples. We plot the average over the 28 pairs of colors.

## B   IMPLEMENTING MISSINGNESS BY DROPPING TOKENS IN VISION TRANSFORMERS

As described in Section 2.2, the token-centric nature of vision transformers enables a more natural implementation of missingness: simply drop the tokens that correspond to the removed image sub-regions. In this section, we provide a more detailed description of dropping tokens, as well as a few implementation considerations.

Recall that a ViT has two stages when processing an input image $\mathbf{x}$.

- **Tokenization:** $\mathbf{x}$ is split into $16 \times 16$ patches and positionally encoded into tokens.
- **Self-Attention:** The set of tokens is passed through several self-attention layers and produces a class label.

After the initial tokenization step, the self-attention layers of the transformer deal solely with sets of tokens, rather than a constructed image. This set is not constrained to a specific size. Thus, after the patches have all be tokenized, we can remove the tokens that correspond to removed regions of the input before passing the reduced set to the self-attention layers. The remaining tokens retain their original positional encodings.

Our attached code includes an implementation of the vision transformer which takes in an optional argument of the indices of tokens to drop. Our implementation can also handle varying token lengths in a batch (we use dummy tokens and then mask the self-attention layers appropriately).

**Dropping tokens for superpixels and other patch sizes**   In the main body of the paper, we deal with $16 \times 16$ image subregions, which aligns nicely with the tokenization of vision transformers. In Appendix C, we consider other patch sizes that do not align along the token boundaries, as well as irregularly shaped superpixels. In these cases, we conservatively drop the token if any portion of the token was supposed to be removed (we thus remove a slightly larger subregion).

# C ADDITIONAL EXPERIMENTS (SECTION 3)

## C.1 ADDITIONAL EXAMPLES OF THE BIAS (SIMILAR TO FIGURE 2).

In Figure 11, we display more examples that qualitatively demonstrate the missingness bias.

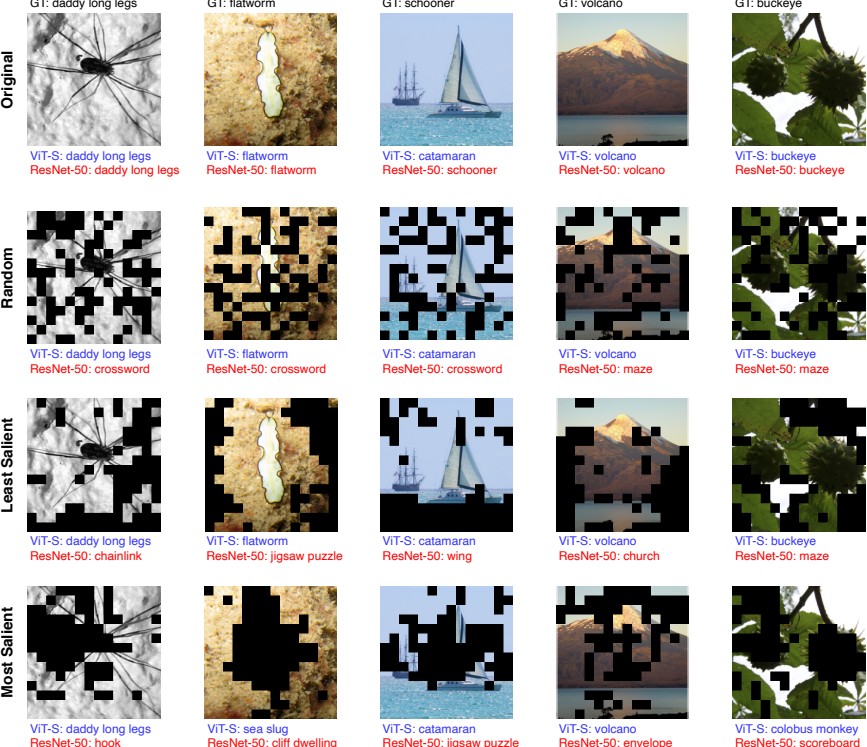

Figure 11: Further examples of removing 75 16 × 16 patches from ImageNet images. The images are blacked out for ResNet-50, and the corresponding tokens are dropped for ViT-S. While ResNet-50 skews toward classes that are unrelated to the remaining image features (i.e crossword, jigsaw puzzle), the ViT-S either maintains its original prediction or predicts a reasonable label given remaining image features.

## C.2 BIAS FOR REMOVING PATCHES IN VARIOUS ORDERS

In this section, we display results for the experiments in Section 3 where we remove patches in 1) random order 2) most salient first and 3) least salient first (Figure 12). We find that missingness approximations skew the output distribution for ResNets regardless of what order we remove the patches. Similarly, we find that the ResNet's predictions flip rapidly in all three cases (though to varying extents). Finally, the ViT mitigates the impact of missingness bias in all three cases.

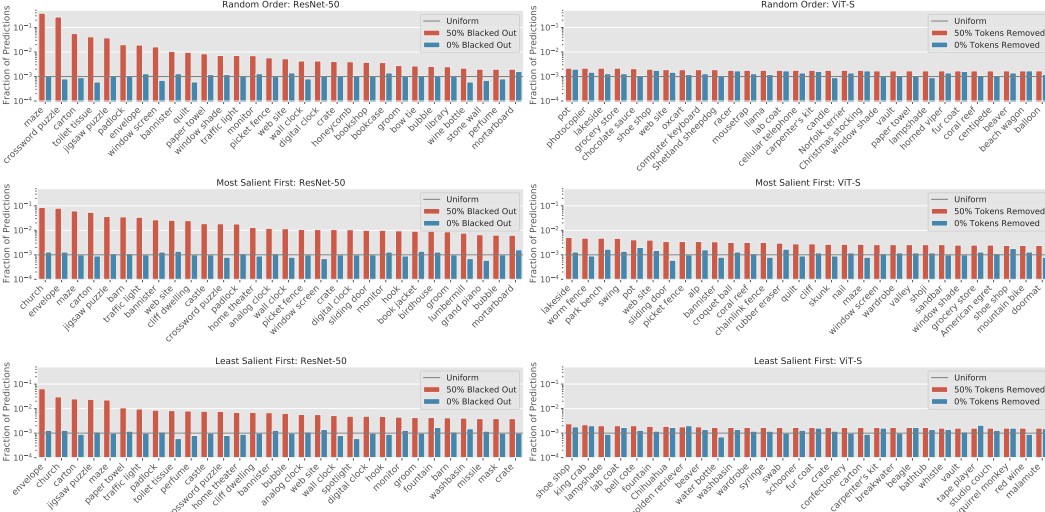

(a) The shift in the output class distribution after applying missingness approximations in different orders.

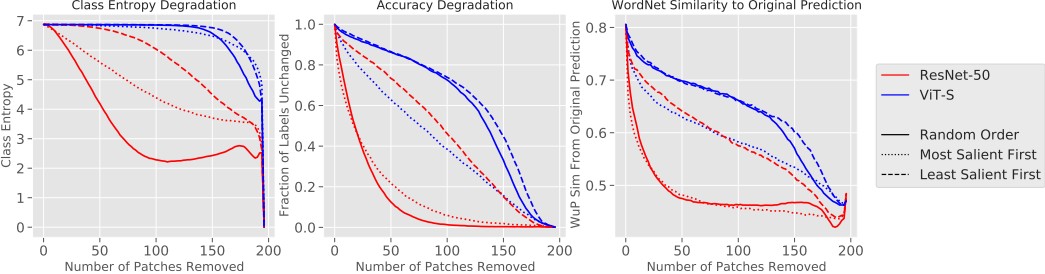

(b) Degradation in class entropy (left), the fraction of predictions that change (middle), and the average Word-Net similarity if the prediction changes after masking (right) as we remove patches from the image in different orders.

Figure 12: Full experiments for removing $16 \times 16$ patches by blacking out (ResNet-50) or dropping tokens (ViT-S).

## C.3 RESULTS FOR DIFFERENT ARCHITECTURES

In this section, we repeat the experiments in Section 3 with several other training schemes and types of architectures. Our results parallel our findings in the main paper.

### C.3.1 VIT-T AND RESNET-18

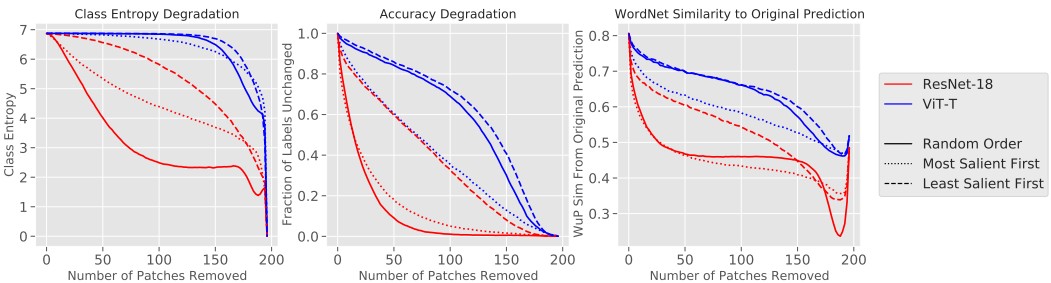

Figure 13: Bias experiments as in Section 3, with a ViT-T and ResNet-18.

### C.3.2 VIT-S AND ROBUST RESNET-50

We consider a ViT-S and an $L2$ adversarially robust ResNet-50 ($\epsilon = 3$).

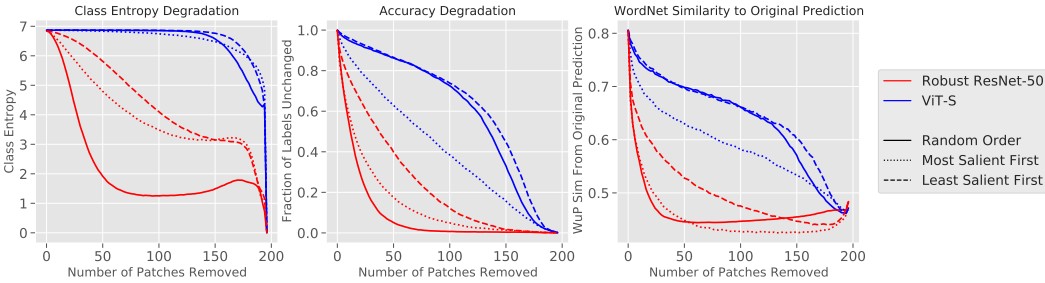

Figure 14: Bias experiments as in Section 3, with a ViT-S and a robust ResNet-50.

### C.3.3 VIT-S AND INCEPTIONV3

We consider a ViT-S and an InceptionV3 (Szegedy et al. (2016)) model.

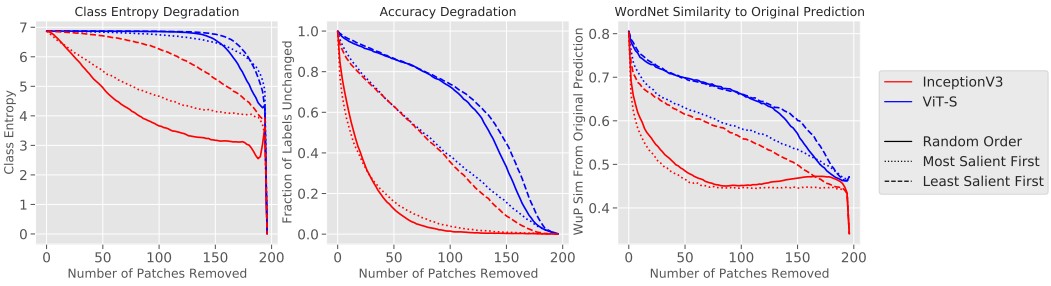

Figure 15: Bias experiments as in Section 3, with a ViT-S and InceptionV3.

### C.3.4 VIT-S AND VGG-16

We consider a ViT-S and a VGG-16 with BatchNorm (Simonyan & Zisserman (2015)).

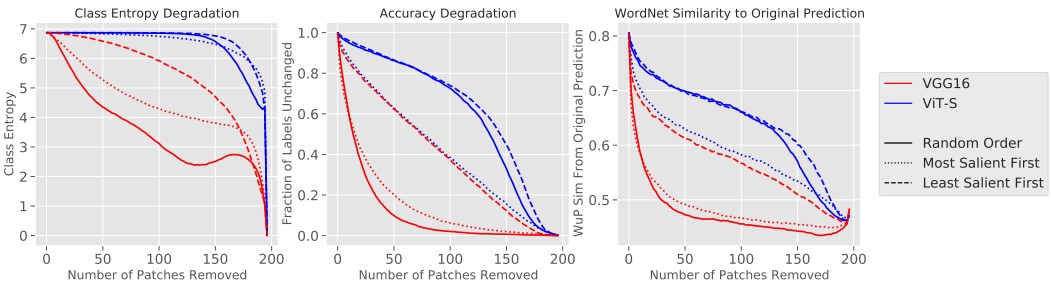

Figure 16: Bias experiments as in Section 3, with a ViT-S and VGG16.

## C.4 RESULTS FOR DIFFERENT MISSINGNESS APPROXIMATIONS

In this section, we consider missingness approximations other than blacking out pixels. In Figure 17, we use three baselines: a) the mean RGB value of the dataset, b) a randomly selected baseline color for each image, and c) a randomly selected color for each pixel Sturmfels et al. (2020); Sundararajan et al. (2017). Since we drop tokens for the vision transformers, changing the baseline color does not change the behavior for the ViTs. Our findings in the main paper for blacking out patches closely mirror the findings for other baselines.

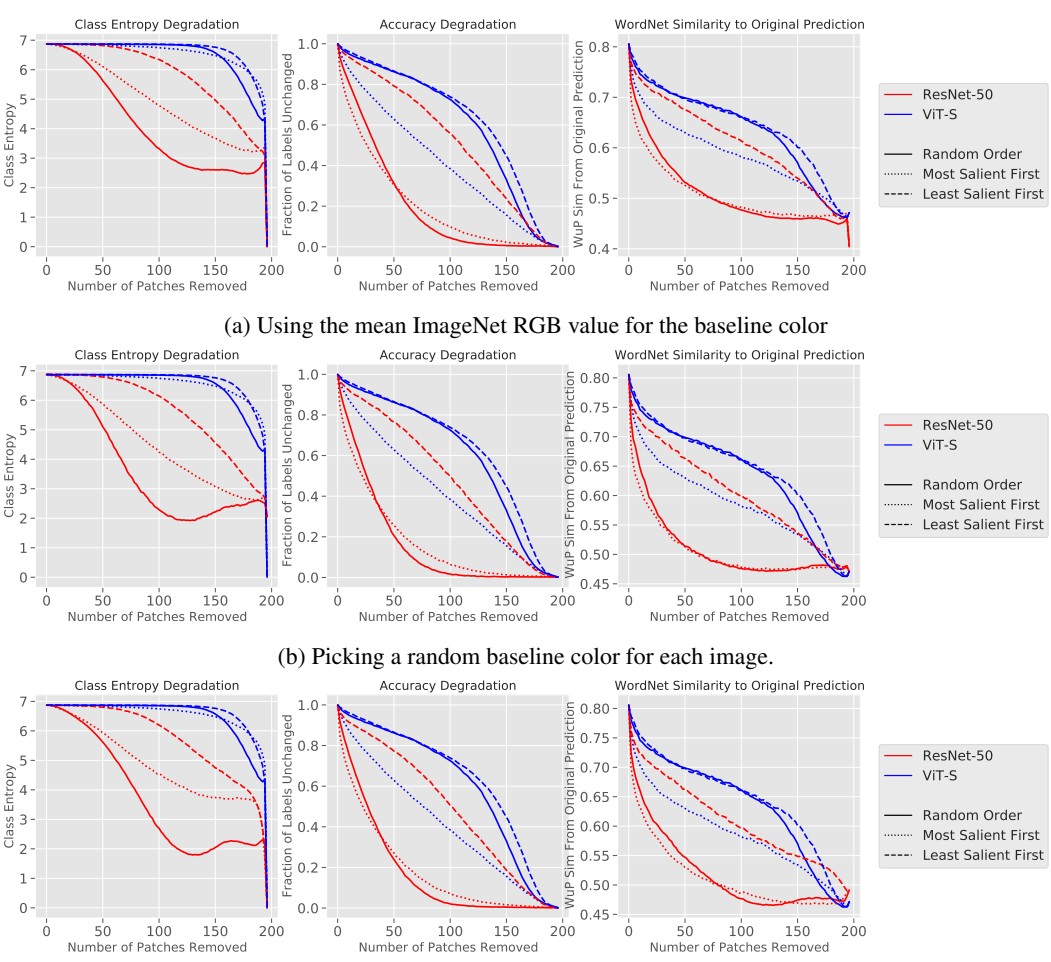

(a) Using the mean ImageNet RGB value for the baseline color

(b) Picking a random baseline color for each image.

(c) Picking a random baseline color for each pixel.

Figure 17: Using different baseline colors for masking pixels.

We also consider blurring the removed features, as suggested in Fong & Vedaldi (2017). We use a gaussian blur with kernel size 21 and $\sigma = 10$. Examples of blurred images can be found in Figure 18a. Unlike the previous missingness approximations, *this method does not fully remove subregions of the input*; thus, blurring the pixels can still leak information from the removed regions, which can then influence the model's prediction. Indeed, we find that, by visual inspection, we can still roughly distinguish the label of images that are entirely blurred (as in Figure 18a).

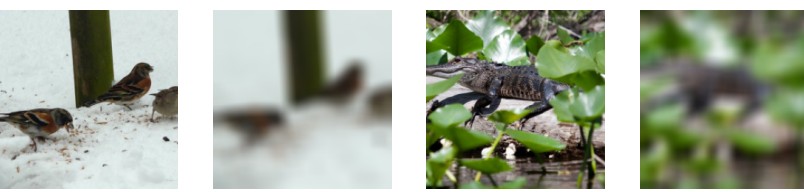

(a) Examples of blurring ImageNet images.

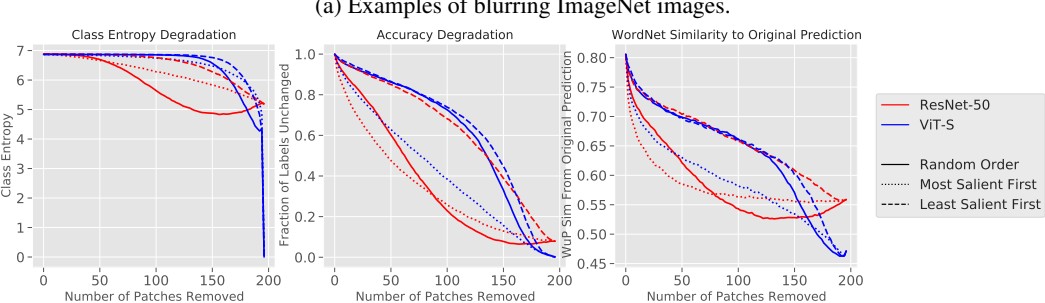

(b) Repeating the experiments in Section 3 using the blurred ImageNet image.

Figure 18: Using the blurred image for the missingness approximation.

For completeness, we repeat the experiments in Section 3 using the blurred image as the missingness approximation (Figure 18b). We find that ResNets still experience missingness bias, though the bias is reduced compared to using an image-independent baseline color.

## C.5 USING DIFFERENTLY SIZED PATCHES

In the main body of the paper, we consider image subregions of $16 \times 16$. In this section, we consider subregions of other patch size: $14 \times 14$, $28 \times 28$, $32 \times 32$, and $56 \times 56$. As mentioned in Appendix Section B, when dropping tokens for the ViT, we conservatively drop the token if *any* part of the corresponding image subregion is being removed. Thus, the ViT removes slightly more area than the ResNet for patch sizes that are not multiples of 16. We find that the ResNet is impacted by missingness bias regardless of the patch size, though the effects of the bias is reduced for very large patch sizes (see Figure 19).

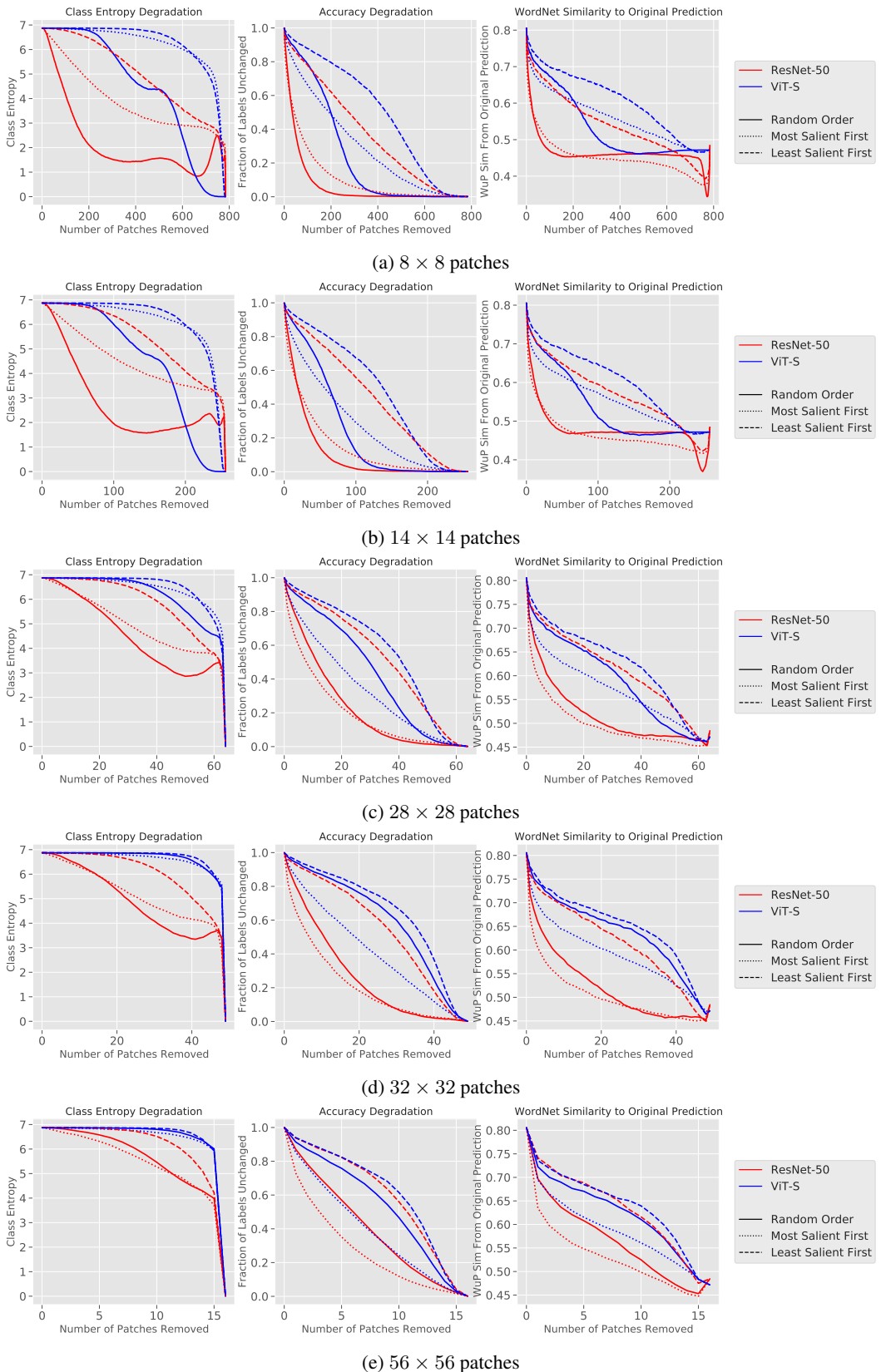

Figure 19: Using different baseline colors for masking pixels.

## C.6 Using superpixels instead of patches

Thus far, we have used square patches. What if we instead use superpixels? In this section, we compute the SLIC segmentation of superpixels (Achanta et al., 2010). In order to keep the superpixels roughly the same size (and of a more similar size as the patches in the paper), we consider the images having more than 130 superpixels. We display examples of the superpixels in Figure 20a.

We then repeat our experiments from Section 3, and analyze our models' predictions after removing 50 superpixels in different orders (blacking out for ResNets and dropping tokens for ViTs). As described in Section B, we conservatively drop all tokens for ViTs that contain any pixels that should have been removed. We find that ResNets are significantly impacted by missingness bias when masking out superpixels. As in the case of patches, dropping tokens through the ViT substantially mitigates the missingness bias.

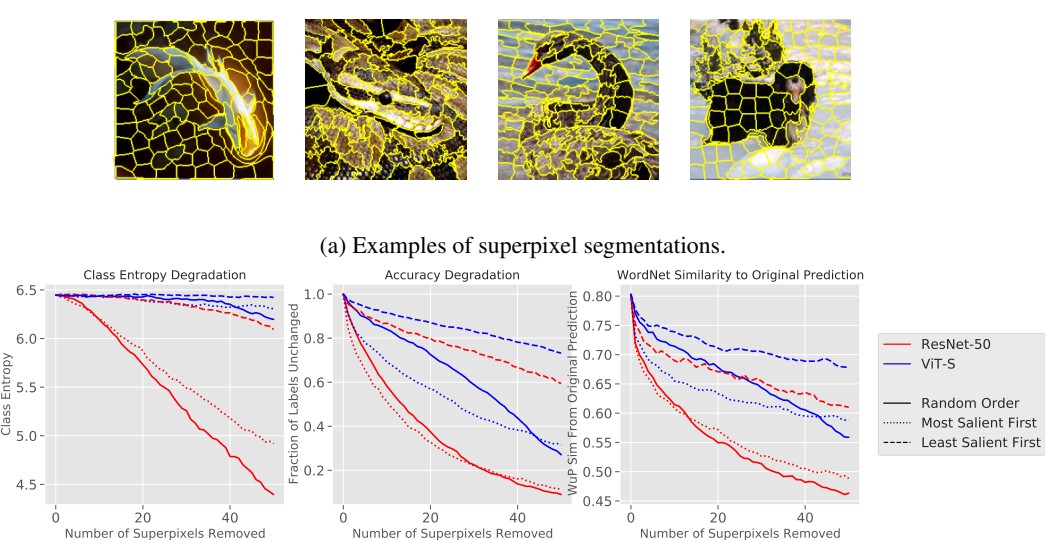

(a) Examples of superpixel segmentations.

(b) Measuring the impact of removing superpixels through missingness approximations.

Figure 20: Using SLIC superpixels instead of patches

## C.7 Comparison of dropping tokens vs blacking out pixels for ViTs

We compare the effect of implementing missingness by dropping tokens to simply blacking out pixels for ViTs. Figure 21, shows a condensed version of the experiments we did previously in this section, but now including an additional baseline which is a ViT-S with blacking out pixels instead of dropping tokens. We find that using either of the ViTs instead of the ResNet significantly mitigates missingness bias on all three metrics. However, dropping tokens for ViTs mitigates missingness bias more effectively than simply blacking out pixels.

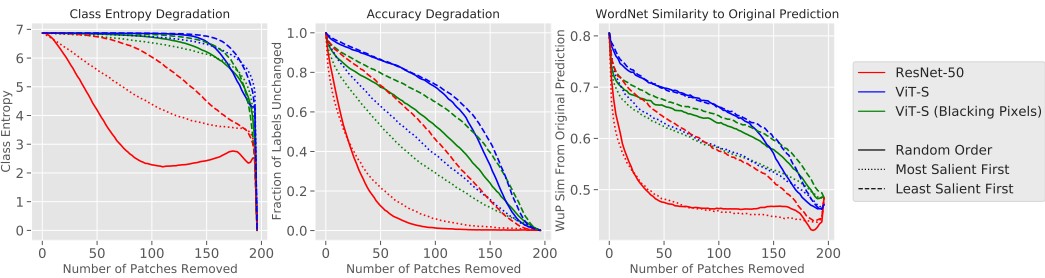

Figure 21: We compare dropping tokens vs blacking out pixels for ViTs.

## D  ADDITIONAL EXPERIMENTS (SECTION 4)

### D.1  EXAMPLES OF LIME

In this section, we display further examples of LIME explanations for ViT-S, ResNet-50, and a ResNet-50 retrained with missingness augmentations (Figure 22). As we explain in Section 4, LIME relies heavily on the notion of missingness, and can be subject to missingness bias. We note that the ViT-S and retrained ResNets qualitatively have more human-aligned LIME explanations (by highlighting patches in the foreground over patches in the background) compared to a standard ResNet. While human-alignment does not a guarantee that the LIME explanation is good (the model might be relying on non-aligned features), we do see a substantial difference in the explanations of models robust to the missingness bias (ViTs and retrained ResNet) and models suffering from this bias (standard ResNet).

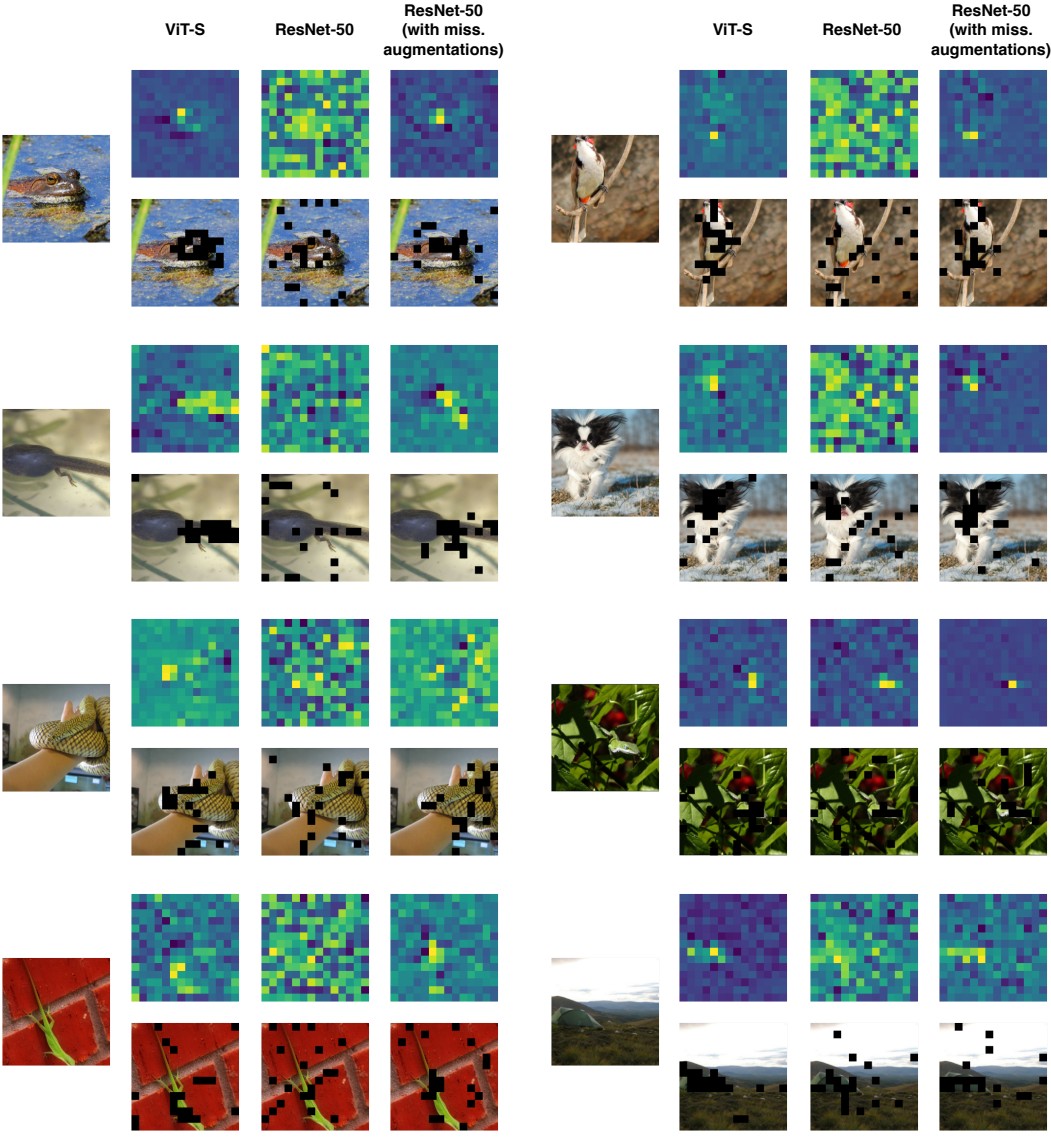

Figure 22: Examples of LIME explanations

## D.2 Top-k ablation test with superpixels.

We repeat the top-k ablation test in Section 4, using superpixels instead of patches (Figure 23). The setup for superpixels is that same as that described in Appendix C.6. After generating LIME explanations for a ViT and ResNet-50, we evaluate these explanations using the top-k ablation test. As we found for $16 \times 16$ patches, the explanations when evaluating with a ResNet are less distinguishable than when evaluating for a ViT: even masking features according to the random explanations rapidly flips the predictions. We do find that masking random superpixels seems to have a greater effect on ViTs than masking random $16 \times 16$ patches: this is likely because the superpixels are on average larger.

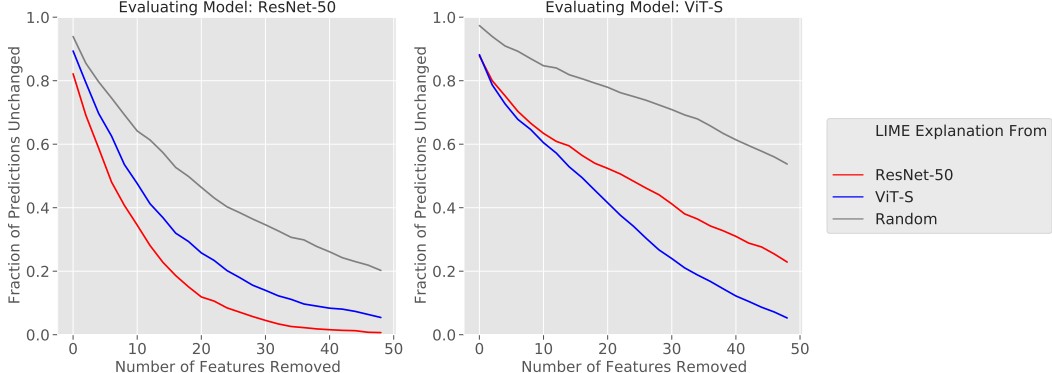

Figure 23: Top K ablation test using superpixels instead of patches.

## D.3 Effects of Missingness Bias on Learned Masks

In this section, we consider a different model debugging method (Fong & Vedaldi (2017)) which also relies on missingness. In this method, a minimal mask is directly optimized for each image. We implement this method at the granularity of $16 \times 16$ patches.

**Model Debugging through a learned mask (Fong & Vedaldi (2017))** Specifically $x$ be the input image, $\mathcal{M}$ be the model, $c$ be the model's original prediction on $x$, and $b$ be a baseline image (in our case, blacked out pixels). The method optimizes for $m$, a $14 \times 14$ grid of weights between 0 and 1 which assigns importance to each patch. We define the perturbation via $m$ as a linear combination of the input image and the baseline image for each patch (plus some normally distributed noise $\epsilon$):

$$f(x, m) = x * \text{upsample}(m) + b * (1 - \text{upsample}(m)) + \epsilon$$

Then the optimal $\hat{m}$ is computed as:

$$\hat{m} = \underset{m}{\arg\min} \, \lambda_1 ||\mathbf{1} - m||_1 + \lambda_2 ||m||_{TV}^{\beta} + [\mathcal{M}(f(x, m))]_c$$

with $\lambda_1 = 0.01, \lambda_2 = 0.2, \beta = 3, \epsilon \sim \mathcal{N}(0, 0.04)$.

The optimal $m$ is computed through backpropagation, and can then be treated as a model explanation. Parameters and implementation were adapted from the implementation at `https://github.com/jacobgil/pytorch-explain-black-box`.

**Assessing the impact of missingness bias.** Since $m$ must be backpropagated, we cannot leverage the drop tokens method for ViTs (which would require $m$ for each patch to be either 0 or 1). However, we generate explanations for the ResNet-50 in order to examine the impact of missingness bias. As in Section 4, we evaluate the explanation generated by this method alongside a random baseline and the LIME explanation for that ResNet using the top-K ablation test (Figure 24). As is the case for the LIME explanations, missingness bias renders the explanations generated by this method indistinguishable from random.

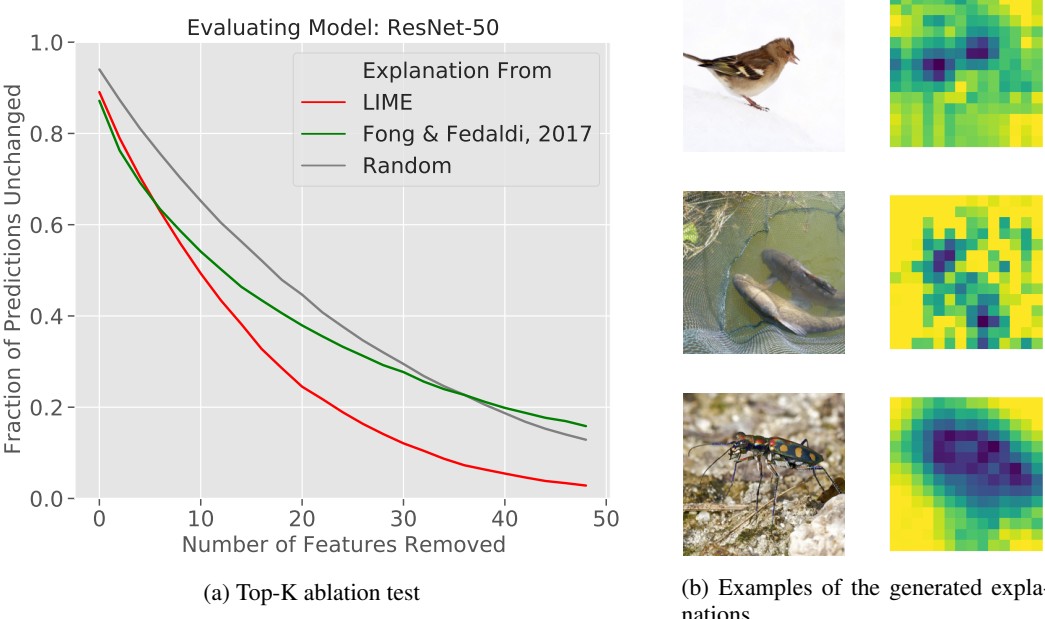

(a) Top-K ablation test

(b) Examples of the generated explanations

Figure 24: (a) Top K ablation test evaluated on a ResNet-50. We evaluate explanations generated by LIME, Fong & Vedaldi (2017), and a random baseline. Due to missingness bias, the explanations are indistinguishable from random (b) Examples of explanations generated by Fong & Vedaldi (2017).

# E  OTHER DATASETS

While our main paper largely considers the setting of ImageNet, we include here a few results on other datasets.

## E.1  MS-COCO

MS-COCO (Lin et al. (2014)) is an object recognition dataset with 80 object recognition categories that provides bounding box annotations for each object. We consider the multi-label task of object tagging, where the model predicts whether each object class is present in the dataset. We train object tagging models using a ResNet-50 and ViT-S, with an Asymmetric Loss as in Ben-Baruch et al. (2020). An object is predicted as "present" if the outputted logit is above some threshold.

We study the setting of removing people from images using missingness. In particular, we seek to remove the image regions contained inside a "person" bounding box that is not contained in a bounding box for another non-person object. Examples of removing people from the image can be found in Figure 25. We then seek to check whether removing the person affected the model predictions for other, non-person object classes.

Specifically, we consider the 21,634 images in the MS-COCO validation set that contain people. For each image, we evaluate our model on the original image and the image with the person removed (blacking out for the ResNet-50 and dropping the token for the ViT). Then, to measure the consistency of the non-person predictions, we compute the Jaccard similarity for the set of predicted objects (excluding the person class) before and after masking. We plot the average similarity over all the images for different prediction thresholds in Figure 25.

We find that the ViT more consistently maintains the predictions of the non-person object classes when masking out people. In contrast, masking out people for the ResNet is more likely to change the predictions for other object classes.

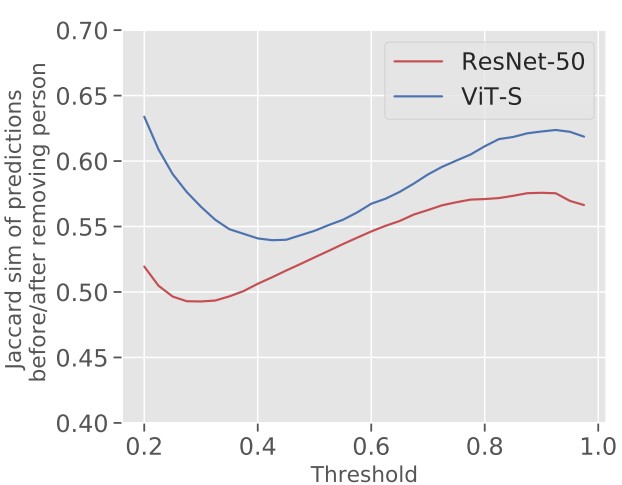

(a) Consistency of non-person prediction after removing people from the images.

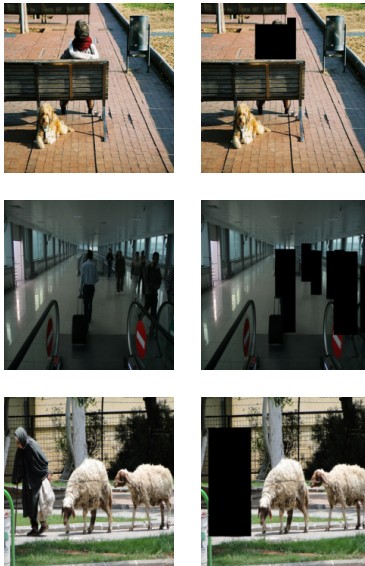

(b) Examples of removing people from MS-COCO images

Figure 25: (a) Average Jaccard similarity of the set of non-person predictions before and after removing all people from the image. We plot over prediction thresholds for the tagging task. (b) Examples of removing people from MS-COCO images.

### E.2 CIFAR-10

We consider the setting of CIFAR-10 (Krizhevsky (2009)). Specifically, we train a ResNet-50 and a ViT-S on the CIFAR-10 dataset upsampled to 224x224 pixels. We start training from the ImageNet checkpoints used throughout this paper. This step is necessary for ensuring high accuracy when training ViTs on CIFAR-10 (Touvron et al. (2020)). We then consider how randomly removing 16x16 patches from the upsampled CIFAR images changes the prediction. Similarly to the case of ImageNet, we find that the ResNet-50 more rapidly changes its prediction as random parts of the image are masked, while the ViT-S maintains its prediction even as large parts of the image are removed.

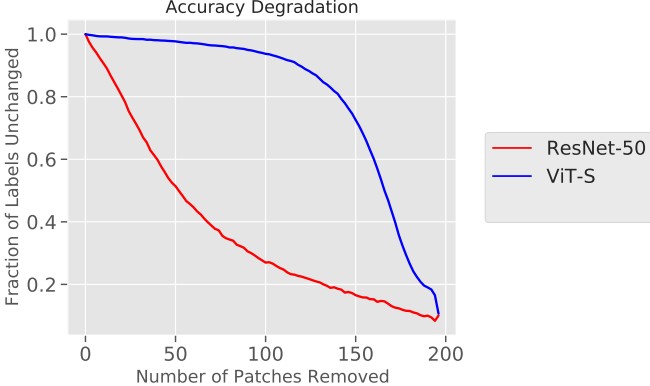

Figure 26: We plot the fraction of images where the prediction does not change as image regions are removed for the CIFAR-10 dataset.

# F RELATIONSHIP TO ROAR

Here, we present more details about the ROAR experiment of Section 3.

## F.1 OVERVIEW ON ROAR

Evaluating feature attribution methods requires the ability to remove features from the input to assess how important these features are to the model's predictions. To do so properly, Hooker et al. (2018) argue that re-training (with removing pixels) is required so that images with removed features stay in-distribution. Their argument holds since machine learning models typically assume that the train and the test data comes from a similar distribution.

So, they propose *RemOve and Retrain (ROAR)* where new models (of the exact same architecture) are *retrained* such that random pixels are blacked out during training. The intuition is that this way, removing pixels do not render images out-of-distribution. Overall, they were able to better assess how much removing information from the image affects the predictions of the model using those retrained surrogate models.

The authors of ROAR list several downsides for their approach though. In particular, retraining models can be computationally expensive. More pressingly, the retrained model is *not* the same model that they analyze, but instead a surrogate with a substantially different training procedure: any feature attribution or model debugging result inferred from the retrained model might not hold for the original model. Given these downsides, is retraining always necessary?

## F.2 VITS DO NOT REQUIRE RETRAINING

Here, we show that retraining is not always necessary: indeed for ViTs, we do not need to retraining to be able to properly evaluate feature attribution methods. While ROAR in (Hooker et al., 2018) dealt with blacking out features on a per-pixel level, we adapt their approach for masking out larger contiguous regions (like patches). If we apply missingness approximations during training as in ROAR, missingness approximations are now in-distribution, and thus would likely mitigate the observed biases.

We retrain a ResNet-50 and a ViT-S by randomly removing 50% of patches during training (through blacking out pixels for the ResNet-50 and dropping tokens for the ViT-S). Our goal is to compare the behavior of each model to its retrained counterpart. If retraining does not change the model's behavior when missingness approximations are applied, retraining would be unnecessary, and we can instead confidently use the original model. In Figure 5, we measure the fraction of images where the model changes its prediction as we remove image features for both the standard and retrained models. We find that, while there is a significant gap in behavior between the standard and retrained CNNs, the standard and retrained ViTs behave largely the same..

This result indicates that, while retraining is important when analyzing CNNs, it is unnecessary for ViTs: we can instead intervene on the original model. We thus avoid the expense of training the augmented models, and can perform feature attribution on the actual model instead of a proxy.

