# OpenReview forum: "Missingness Bias in Model Debugging"
_ICLR.cc/2022/Conference — ICLR 2022 Poster_

### Official Review · Reviewer_hiFn · 2021-11-02

**Correctness:** 4
**Technical Novelty And Significance:** 2
**Empirical Novelty And Significance:** 3
**Recommendation:** 5
**Confidence:** 5

**Main Review:**

Strengths:
i) The authors have analyzed a very relevant problem in vision, especially explainable AI as applied in vision. Perturbing inputs (images) can be a strong tool towards finding importance of regions of input, however the authors showed through a number of well designed experiments and evaluation metrics that, if wrongly, applied this can be harmful too.
ii) The experiments are well thought and executed. A natural progress is seen in going about the experiments starting with trying to find if masking image regions make the classifier look for only a few classes of images, flipping class decision is good if important regions are masked, but whether class decisions are flipped even with random or unimportant region removal, taking LIME as a case study, the authors also tried to see if changing baseline colors in the mask region results in consistent explanations and so on.
iii) This work definitely does open more doors into research of better debugging tools rather than removing regions from images.
iv) The paper is clearly written.

Weaknesses:
i) The authors have shown all their results in section 3 by only blackenning rather than any other baseline value. Blackenning is majorly known to be an incorrect replacement for the removed pixels [a, b]. Other common methods are blurring, graying (using the mean values) or random noise. It would be interesting to see if at all these baselines also show the same effects of missingness bias.
ii) The authors have used LIME as the case study for generating explanations [sec 4]. Some relevant approaches (e.g., [a, b, c]) provide much smoother saliency maps which might be worth trying especially to check inconsistency and indistinguishability of explanations as well as the effect dependence on maze/crossword like pattern [Figs. 2 and 3].
iii) It would have been good if authors could show the results on more models like EfficientNet or even VGG instead of only ResNet. This would show that this problem exists in CNNs in general and is not specific to only ResNet.
iv) (Fig. 8) Does it make sense to find a measure of agreement for different baseline colors in ViT. These baseline colors are not used at all while making ViT prediction for these masked images. So, it is quite natural that all these so called ViT variations will give exactly the same features with same importance for all these baseline methods. Also it would be good to know what are the 8 different baseline colors?
v) Instead of being an weakness, this is rather an academic query: There is no major alternative to the current method followed for removal of regions in images for CNNs. This problem is not talked of much.  After analyzing an already existing problem, it would have been great if the authors would have talked about any method to overcome it. Shifting to transformer based architecture may not be a solution for explaining and debugging CNNs.

[a] Petsiuk et. al., RISE: Randomized Input Sampling for Explanation of Black-Box Models, BMVC 2018
[b] Fong et. al., Interpretable Explanations of Black Boxes by Meaningful Perturbations, ICCV 2017
[c] Fong et. al., Understanding Deep Networks via Extremal Perturbations and Smooth Masks, ICCV 2019

**Summary Of The Paper:**

The paper discusses that it is common in Computer Vision debugging and explainability techniques to remove image regions to attribute different regions of the image to the decision of a classification model. Although such removal (of words) can be beneficial for Natural Language model debugging, it adds an additional bias in Computer Vision. This is because removing regions implies replacing the corresponding pixels with some baseline values e.g., black color, random intensities, average pixel values etc. The paper shows that irrespective of which part of the region is being masked (i.e., removing original image pixel values and replacing with baseline values), masking small portions of image can lead to CNNs predicting incorrectly and unreliably. The authors show that the CNN based classifiers seem to rely on the “masking pattern” to make the prediction, rather than the remaining (unmasked) portions of the image. In fact, even after removing some image regions randomly, the output distribution gets highly skewed towards a few classes e.g., crossword, jigsaw puzzle etc. Using LIME as a case study, it shows that missingness bias can lead to inconsistent and indistinguishable explanations. Moreover, this bias can be overcome if the model is trained with suitable augmentations that remove regions while training.
The paper illustrates how using Visual Image Transformers is a better natural choice as these models allow actual removal of image regions rather than replacement with baseline values. Hence, these issues seen in CNNs due to missingness bias is not prevalent in ViTs.

**Summary Of The Review:**

Overall, this work is good showing crisp thinking and analysis of the experiment design. However, few more experimentation would have been better to have. Some such experiments are listed in detail in the weaknesses section above. The approximation of “missingness of pixels” by supposed to be “meaningless pixels” is a longstanding problem in vision and researchers are aware of this. This work definitely will help progress the research in this. However, I would like to see the authors to discuss some possible solutions to this problem so that the work does not look like an illustration of an already known problem in Computer Vision only.

---

> ### Author Response · Authors · 2021-11-16
> **Response to Reviewer hiFn**
>
> We thank the reviewer for their comments. We address the specific concerns below:
>
> **[Additional approximations of missingness]** We would like to point the reviewer to Appendix C.5 which contains results for other baselines that the reviewer brought up (using the mean pixel value, using a random color per pixel/image, blurring the image). These results are all consistent with our main findings, and so we deferred them to the appendix.
>
> **[Considering other model debugging techniques]** We thank the reviewer for their suggestions on other explainability methods. In the new revision, we have implemented one of the suggested techniques (https://arxiv.org/pdf/1704.03296.pdf) and present the effect of missingness bias in Appendix D.3. We show that missingness bias causes this type of  explanation to be  indistinguishable from random explanations, consistent with our main result on LIME.
>
> **[Additional architectures]** We have added results in the new revision for more architectures (InceptionV3 and VGG-16) in Appendix C.4.3 and C.4.4. We find that similar results to ResNets hold.
>
> **[Baseline colors for inconsistency experiment]** The goal of this experiment was to highlight how *ResNets* are very sensitive to the specific missingness approximation that is used. Since ViTs are consistent by construction, we include them mainly as a bias-free comparison point for the ResNets. The interesting takeaway is that explanations for ResNets can change quite a bit depending on the baseline color, which is quite concerning.
> We refer the reviewer to Appendix A.4 for details about the experiment and the specific colors used (in brief, the 8 colors are generated by setting the R, G, and B values to either 0 or 1. This gives 2^3=8 different colors).
>
> **[Solution for explaining and debugging CNNs]** We thank the reviewer for raising this question. It is indeed important to know whether there are solutions for solving this bias problem for CNN in situations where a CNN can’t be replaced with a ViT. As we discuss in our response to Reviewer SQBL, directly dropping masked pixels is nontrivial for CNNs due to the growing receptive field.
>
> One potential solution to mitigate bias for CNNs is training with missingness augmentations. We discussed and evaluated this at the end of S4, and in more detail in Appendix E. This approach has trade-offs however: retraining can be very expensive and can lead to drops in clean accuracy.
>
> **Relevant Revisions:** In the new revision, we added:
> - Additional experiment for model debugging technique from Fong and Vedaldi, 2017 (Appendix D.3)
> - Additional experiments for InceptionV3 and VGG-16 (Appendix C.4.3 and C.4.4)

---

### Official Review · Reviewer_SQBL · 2021-11-02

**Correctness:** 3
**Technical Novelty And Significance:** 2
**Empirical Novelty And Significance:** 3
**Recommendation:** 5
**Confidence:** 4

**Main Review:**

The motivation of this paper is interesting, the paper is well-written and it uses various visualizations to demonstrate the issue on CNNs/ResNets. However, my concerns are the following:

-  The paper claims that a ‘hole’ cannot be left in the image in convolutional networks. This is not clear why it is the case, since a convolutional network uses a sliding window to convolve each pixel with the kernel of the layer. Then, each output in the feature map is exactly the output that corresponds to a specific positioning of the kernel. So, similarly to the way that we can ‘leave out’ patches of the image, we could implement the sliding window to skip some patches, so that the feature map (and every next feature map) would be exactly zero in those dimensions.

- The previous point of skipping some parts of the image with the sliding window, makes me wonder what the results on ResNet would be in this case; currently, the paper showcases poor results on ResNet, but it seems plausible that it was not optimized for this task.

- The paper experiments with patches that are only rectangular and size of 14x14 or bigger (appendix C.6). How would it work in the case of non-rectangular patches and or small ones?

- I am not convinced about the **utility of the proposed method**; ViT is a state-of-the-art method only when it comes to extremely large-scale image recognition; in fact, the authors of ViT recognize that if only trained on datasets like ImageNet it can perform similarly to ResNet. In other words, to learn ViT, we need a huge annotated dataset, which practically means that it can be used for debugging imagenet (or a similar dataset) but it is hard to be used for any other dataset. This calls into question whether it can be a general tool, or something useful only for a handful of ImageNet-like datasets. For instance, **can the proposed tool be used for debugging classifiers on medical imaging or similar constrained-size datasets** that are not related to ImageNet?

- Another limitation of the method is that practically only a single task and mainly ImageNet (?) datasets are used for the evaluation. The contribution is based on the empirical validation across a board of tasks and across datasets, which at the moment is not the case. How can the proposed method generalize to different datasets, e.g., ChestRay or food101 datasets?

- Currently, only ResNet is used as a baseline, while previous work [Sturmfels, 2020] used Inception v4 instead. Also, EfficientNet or similar networks have recently been performing well in terms of image recognition. It would have been better to showcase that those also suffer from the same issue as ResNet.


**Summary Of The Paper:**

The authors focus on the problem of model debugging (for image recognition). They identify that the proposed tools (that rely on CNNs and ResNets) might suffer from the ‘missingness’ issue, i.e., the absence of features due to masking objects of interest. The authors exhibit how the method of masking pixels/patches can lead to the missingness bias and this can happen even if the masked values are replaced with some other 'dummy' values. To mitigate that, they propose to use a recent transformer since it does not use convolutions, but linearized patches as inputs. Then, they demonstrate that the proposed method can simply 'skip' those patches that are masked and it does not result in a skewed result.


**Summary Of The Review:**

The paper makes an observation on the absence of features for debugging purposes and argues that ViT should be preferred over standard CNNs. The paper exhibits the use case on ImageNet, but it does not demonstrate how this can be useful for diverse datasets, or even the utility of the method in various cases (e.g. non-rectangular, small patches) and tasks.

---

> ### Author Response · Authors · 2021-11-16
> **Response to Reviewer SQBL**
>
> We thank the reviewer for their comments. We address their specific concerns below:
>
> **[Sliding window to skip patches for CNNs]** The suggested method for dropping pixels (and the corresponding regions in the feature map) could potentially work for CNNs if we have only 1 convolutional layer. But with multiple convolutional layers, the masked out region would “propagate” and grow larger and larger after each convolutional layer. In this case, the growing receptive field for CNNs becomes a problem: any removed region will increase in size and can eventually remove the entire feature map if the network is deep enough. Thus, dropping pixels for a CNN is not so straightforward, but could be an interesting avenue for future work. Note that ViTs do not have a growing receptive field and hence do not have this problem.
>
> **[Non-rectangular patches]** We would like to point the reviewer to Appendix C.7 and D.2 where we included results for using superpixels instead of patches for the missingness analysis. As per the reviewers suggestion, we also add analysis of even smaller patches (i.e 8x8) in Appendix C.6.
>
>
> **[ViTs need huge amounts of data, does our method work on smaller datasets]** Yes. Although ViTs do perform best with larger amounts of data, it is known that pre-trained ViTs can be finetuned on small datasets to achieve very high accuracy. For example, see Table 6 of https://arxiv.org/pdf/2012.12877.pdf, which does this for six smaller datasets (as small as 2k examples). With pre-training, we can train and debug ViT models on small datasets (see next bullet point for additional datasets that we have since added).
>
> **[Additional datasets]** In the new revision, we add further experiments studying the missingness bias for the MS-COCO and CIFAR-10 datasets (see Appendix E). These results are consistent with our main results.
>
> **[Additional architectures]** As per the reviewer’s suggestion, we conducted our analysis on more architectures (specifically InceptionV3 and VGG-16), and we obtained similar results to ResNets. Please see Appendix C.4.3 and C.4.4.
>
> **Relevant Revisions:** In the new revision, we added:
> + Analysis of missingness bias when using 8x8 patches (C.6)
> + Additional experiments for MS-COCO and CIFAR-10 (Appendix E)
> + Additional experiments for InceptionV3 and VGG-16 (Appendix C.4.3 and C.4.4)

---

> > ### Comment · Reviewer_SQBL · 2021-11-20
> > **Clarification questions**
> >
> > The authors have answered some of the original questions raised in my review. To clarify further, please find below some follow-up questions:
> >
> > * What is the training time of the ViT? What about the computational overhead? What are the requirements (FLOPs, GPUs, time) for running this method? The paper mentions an internal cluster with multiple GPUs, but it is not clear to me what is the minimum requirement one could run this model and the computational time for this with respect to older alternatives.
> >
> > * The revised manuscript conducts experiment on CIFAR10. However, CIFAR10 is a subset of imagenet, so finetuning there should be consistent. This still does not mean it is a useful tool for diverse datasets.
> >
> > *  I am not sure the experimentation with super pixels is clear to me. One of the core ideas here was that by not feeding in patches that are black, you can obtain superior performance. However, as the images demonstrate, super pixels do not have such a canonical form (e.g. rectangular). The paper currently mentions 'irregularly shaped superpixels. In these cases, we conservatively drop the token if any portion of the token was supposed to be removed (we thus remove a slightly larger subregion).'. Doesn't this introduce a bias as well by dropping a superset of the pixels?

---

> > > ### Author Response · Authors · 2021-11-21
> > > **Response to clarification questions**
> > >
> > > We thank the reviewer for taking the time to go over our reply. We address the followup questions below.
> > >
> > > **[Computational time for ViT]:**
> > > We note that ViTs come in various sizes (e.g. ViT-T and ViT-S) that have similar computational requirements as commonly used ResNets (e.g. ResNet-18 and ResNet-50). So, in general, the computational requirements for training and using ViTs are not significantly different from standard ResNets. We include the specific computational times on our hardware below:
> > >
> > > + For ImageNet, we train all our ViTs and ResNet models on 4 V100s each. Training ViT-T is about the same for a ResNet-18 (around 12 hrs), and that of ViT-S about the same as ResNet-50 (around 20 hrs).
> > > + For CIFAR-10, we fine-tune pretrained ViTs and ResNets on 1 V100. Fine-tuning ViT-T and ResNet-18 takes ~ 1 hr, and fine-tuning ViT-S and ResNet-50 takes ~1.5 hrs.
> > >
> > > All of our analysis can be run on a single 1080 Ti GPU, where a forward pass with batch size of 128 takes:
> > > - ResNet-18: 0.033 +- 0.013 sec
> > > - ViT-T: 0.031 +- 0.018 sec
> > > - ResNet-50: 0.039 +- 0.015 sec
> > > - ViT-S: 0.041 +- 0.016 sec
> > >
> > > As you can see, using a ViT instead of a ResNet does not incur substantially different computational costs. We have updated the revision to include this information in Appendix A.
> > >
> > > **[More datasets]:**
> > > We would like to bring the reviewer’s attention to our results on MS-COCO in Appendix E. MS-COCO is a substantially different dataset than ImageNet: the pictures are scene-based and there are multiple objects per scene.
> > >
> > > Also, we kindly note that CIFAR-10 is in fact *not* a subset of ImageNet. CIFAR-10 is a subset of TinyImages (e.g. see https://www.cs.toronto.edu/~kriz/cifar.html), which is a different dataset from ImageNet. We ran the experiments on it to show that the results hold on smaller datasets as was previously requested.
> > >
> > > **[Bias when dropping a superset of superpixels with ViTs]:** It is indeed possible that dropping a slightly larger superset of superpixels might incur some bias due to the loss of information. Even in this setting, our experiments show that ViTs have significantly less missingness bias than ResNets (e.g. the experiments in C.7). Note that in general, a single dropped token contains only 0.5% of the total number of pixels, and so in practice this approach does not lose much information.

---

> > > > ### Comment · Reviewer_SQBL · 2021-11-26
> > > > **Follow-up remarks**
> > > >
> > > > I am thankful for the remarks and the technical correction on CIFAR10. However, this does still not solve my original review remark: the updated manuscript conducts experiments on CIFAR10 (subset of Tiny-imagenet, which is identifies as a [similar challenge](http://cs231n.stanford.edu/reports/2017/pdfs/930.pdf)), and MS-COCO that has images from natural scenes.
> > > > However, tasks that have less data and are not on natural scenes (e.g. medical imaging) might perform differently with the proposed debugging tools.
> > > >
> > > > Additionally, I am still not sure how masking out the pixels in a CNN cannot work in multiple layers. There is a sliding window with a kernel, which you can select *not* to 'convolve' some area with the kernel. It might be the case that it does not propagate, but at the moment the responses mention that somehow it does not propagate without providing a clear reasoning why this would be the case.
> > > >
> > > > As for the parameters, the smallest ViT model (Table 1 [here](https://arxiv.org/pdf/2010.11929.pdf)) seems to be using 86 million, versus the 5 million/22 million used in this work. It would be beneficial to explain the differences (at least in Appendix A that currently does not mention them).

---

> > > > > ### Author Response · Authors · 2021-11-26
> > > > > **Additional clarification**
> > > > >
> > > > > We thank the reviewer for following up and discussing our submission with us, and clarify these remaining points:
> > > > >
> > > > > **[Tasks with less data or not on natural scenes]** We are happy to include results on even more datasets in the final revision, such as the ChestRay or Food101 datasets. We do note, however, that these datasets are not substantially different from what we have already included. For example, in terms of *tasks with less data*, both ChestRay and Food101 consist of 112k and 100k training images, which is similar in size to MS-COCO (we used the 2014 release which has 82K training images) and actually larger than CIFAR10 (50k training images). In terms of tasks that are not on *natural scenes*, both CIFAR10 and ImageNet are already not scene-based datasets. We can certainly add similar analysis on a medical application such as ChestRay in the final revision.
> > > > >
> > > > > For historical accuracy, we note that the reviewer seems to have confused the 80 Million Tiny Images dataset with the Tiny ImageNet dataset. CIFAR10 is a subset of 80 Million Tiny Images (as noted on the CIFAR10 home page https://www.cs.toronto.edu/~kriz/cifar.html), which is *not* a subset of ImageNet and was directly scraped from the internet (https://people.csail.mit.edu/torralba/publications/80millionImages.pdf).
> > > > >
> > > > > **[Clarification for skipping regions for CNNs]** In order to select not to convolve an area with a convolutional kernel, there are two direct ways to do this which we describe in further detail.
> > > > >
> > > > > 1. For each particular sliding window, we can choose to only convolve a subset of the pixels by only using a subset of the kernel in order to ignore masked pixels. This is mathematically equivalent to setting the masked pixel to 0 (i.e. blacking out), and is thus a baseline that we already consider that still induces bias.
> > > > > 2. Alternatively, we can instead drop entire windows if they contain masked regions. This is analogous to our approach of dropping tokens for ViTs, and what we described in our original response.
> > > > >
> > > > > For the latter approach, if we do not want to convolve any areas that overlap with the missing region, we also must skip the border surrounding the missing region. This causes the region that needs to be ignored to grow with each successive convolution.
> > > > >
> > > > > As a concrete example, consider the beginning layers of a typical ResNet-50 (ignoring ReLU’s, BatchNorms, and 1x1 convolutions since they do not increase the receptive field):
> > > > >
> > > > > + 7x7 Convolution, Stride 2
> > > > > + 3x3 MaxPool, Stride 2
> > > > > + “Layer 1”
> > > > >     + 3x3 Convolution, Stride 1
> > > > >     + 3x3 Convolution, Stride 1
> > > > >     + 3x3 Convolution, Stride 1
> > > > >
> > > > > Each convolutional or max pooling layer of size KxK will increase the size of a missing region by 2*(K-1) in both dimensions. If we start with a missing region of size 4x4 and drop sliding windows with missing data, the missing region will grow to 16x16 after the first convolution, and 32x32 by the end of Layer 1. As we move through the additional layers, this “dead zone” continues to grow and eventually the entire feature map becomes a missing region, which is unusable.
> > > > >
> > > > > **[Clarification on ViT sizes]** The sizes in the referenced table range from ViT base to ViT Huge. In our work, we use ViT small and ViT tiny, since they mirror the sizes of a ResNet-50 and ResNet-18. The parameter counts can be found in the official repository of the DeIT paper (https://github.com/facebookresearch/deit). The differences are explained in this paper (https://arxiv.org/pdf/2012.12877.pdf), where these model sizes were introduced explicitly as a counterpart to ResNet-18 and ResNet-50.

---

> > > > > > ### Comment · Reviewer_SQBL · 2021-11-29
> > > > > > **Response**
> > > > > >
> > > > > > I am thankful to the reviewers for the answer. I will keep my score the same, as I believe some experiments with different statistics (e.g. medical imaging ones) are important. In addition, a recommendation for the revised manuscript would be to make it clearer to the reader what the differences in implementation are, since for me I needed to go back and forth between the submitted work and ViT to understand the parameters, the running times (that differ from ViT) etc.
> > > > > >
> > > > > > As a side-note: I am not particularly fixed in ChestRay or Food101 datasets, the point from the original review was datasets that can be useful in different domains; probably there are several other datasets that fulfil this criterion.

---

### Official Review · Reviewer_sYZg · 2021-11-03

**Correctness:** 3
**Technical Novelty And Significance:** 3
**Empirical Novelty And Significance:** 3
**Recommendation:** 6
**Confidence:** 4

**Main Review:**

Overall the paper is well written and the authors provide clear explanations at each stage whilst analyzing the effect of various missingness methods across convolutional neural networks and the Vision Transformer architecture. The authors build on the compelling case that when using debugging methods such as LIME, that iteratively block out features (or regions of the input image in the vision case) to analyze the behavior of convolutional neural networks with and without those features, the bias introduced by the blocked out regions (via semi-arbitrary manipulation such as blackout, blurring, etc) can dramatically impact the behavior of the model. As a solution to this problem, the authors analyze (and compare with CNNs) the use of a VIT architecture to simply drop target regions instead of blocking them out. For VIT architectures, dropping target regions is a clever solution that was demonstrated to produce superior explainability and debugging results when compared with a CNN and blackout or blurring.

While overall I found the paper to provide meaningful insight and analysis, the paper could be improved through further analysis of adversarial robustness of CNNs vs VITs. CNNs have been demonstrated to overfit non robust features in the feature space (features that allow the classifier to achieve high accuracy but have no or little relation to the object class), whereas VITs have as of late been shown to be more robust to such noise. While some analysis was conducted when analyzing the retraining of models with missingness augmentation, given that retraining proved to significantly improve the results from the CNN, I would encourage the authors to more thoroughly analyze whether the superior capability of the proposed VIT token drop method is a result of this behavior rather than blackout and blurring being inferior/problematic methods.

**Summary Of The Paper:**

This paper analyzes the bias introduced when regions of images are modified in order to measure the effect of a model with and without those features present in the image, for model debugging purposes. The authors compare different removal methods, and subsequently analyze how a Vision Transformer architecture might be used instead of a CNN to actually drop features rather than artificially removing them.

**Summary Of The Review:**

Overall the work provides meaningful analysis and insight into the use of debugging methods such as LIME with CNN and VIT architectures, and shows the superiority of token dropping in VIT to blackout, blurring or other feature removal methods in CNNs. Additional analysis by the authors into feature robustness and retraining with CNNs vs VIT to determine whether the superior explainability is due to the regions not being considered in the token drop method as argued by the paper, or whether the propensity for more robust features in VITs is the reason.

---

> ### Author Response · Authors · 2021-11-16
> **Response to Reviewer sYZg**
>
> We thank the reviewer for their comments. We address their specific concerns below:
>
> **[Adversarial robustness analysis]** Whether ViTs are naturally more robust than CNNs is a question that is still contested: while earlier papers indicated that ViTs may be more robust, a recent paper (https://arxiv.org/abs/2111.05464) found that, when keeping training details such as data augmentation schemes constant, CNNs and ViTs are similarly robust. We made sure to keep training settings such as data augmentation the same to provide a fair comparison in our analysis.
>
> Following the reviewers suggestion, we added an experiment in the new revision to study whether missingness bias similarly affects an adversarially robust ResNet-50 (See Appendix C.4.2 in the revision). We find that the missingness bias still exists, indicating that removing reliance on non-robust features does not solve the problem for CNNs.
>
> **[Dropping Tokens vs. Blacking out for ViTs]** We did this experiment in Appendix C.3, where we compared the impact of missingness bias for dropping tokens vs blacking out patches. ViTs incur less bias than ResNets when blacking out pixels, indicating that the architecture change can mitigate this issue to some extent. However, dropping tokens incurs even less bias than blacking out pixels for ViTs. Thus at least part of the missingness bias is due to blacking/blurring. Together, using ViTs and dropping tokens is able to almost fully side-step the missingness biases that ResNets face, and is thus the main focus of the paper.
>
> **Relevant Revisions:** Added experiment for a robust ResNet-50 (Appendix C.4.2)

---

### Author Response · Authors · 2021-11-16
**Revisions to Paper**

We thank the reviewers for their feedback. We have uploaded a revised pdf of the paper, which contains the following additions as requested:
- Bias analysis of an adversarially robust ResNet (Appendix C.4.2)
- Bias analysis of InceptionV3 and VGG16 (Appendix C.4.3, C.4.4)
- Bias analysis of using 8x8 patches (Appendix C.6)
- Additional experiments for understanding the impact of missingness bias for the model debugging technique from [Fong and Vedaldi, 2017](https://arxiv.org/abs/1704.03296) (Appendix D.3)
- Additional experiments on MS-COCO and CIFAR-10 (Appendix E)

These additional results are consistent with our findings in the main body of the paper.

---

### Decision · Program_Chairs · 2022-01-20

**Decision:**

Accept (Poster)

**Comment:**

This work identifies an interesting bias that can occur when applying occlusion based interpretability methods to debug image classifiers. For context, the motivation behind many of these methods is that by occluding various parts of the image, one can ask counterfactuals such as "what would the model have predicted if this object were not present in the image"? However, the authors note that when occluding pixels, classifiers are still functions of the occlusions themselves, so this process may introduce a bias as a result. This is most clearly demonstrated in Figure 2 where a convolutional architecture classifies various occluded images as "jigsaw" or "crossword puzzle", arguably due to the fact that scattered patch based occlusions resemble crossword puzzles. The authors then demonstrate that ViT models can be modified in a way to ask the above counter-factual in a more principled manner---namely by dropping image tokens within the transformer model, the resulting function doesn't take any occluded pixels as input. Reviewers all found the analysis quite insightful, and did not find any significant flaws in the experiments. During the rebuttal, the authors added numerous experiments to address concerns raised by reviewers regarding lack of datasets for which the method was run on. Unfortunately, only one of the reviewers acknowledged the rebuttal and did not raise their score citing doubts that the method may not work well on datasets with differing image statistics (e.g. medical imaging). After reading all of the reviews and rebuttal, the AC feels the authors have adequately addressed the most pressing reviewer concerns, and finds the presented analysis sufficient to warrant acceptance.